# Does Aiming for Long-Term Non-Decreasing Flow of Timber Secure Carbon Accumulation: A Lithuanian Forestry Case

**Gintautas Mozgeris [1],\*** 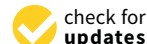**, Vaiva Kazanavičiūtė [2] and Daiva Juknelienė [1]**

1. Agriculture Academy, Vytautas Magnus University, Studentų Str. 11, 53361 Akademija, Lithuania; daiva.jukneliene@vdu.lt
2. Department of Environmental Sciences, Faculty of Natural Sciences, Vytautas Magnus University, Universiteto Str. 10-314, 53361 Akademija, Lithuania; vaiva.kazanaviciute@stud.vdu.lt
* Correspondence: gintautas.mozgeris@vdu.lt

**Abstract:** Lithuanian forestry has long been shaped by the classical normal forest theory, aiming for even long-term flow of timber, and the aspiration to preserve domestic forest resources, leading to very conservative forest management. With radically changing forest management conditions, climate change mitigation efforts suggest increasing timber demands in the future. The main research question asked in this study addresses whether current forest management principles in Lithuania can secure non-decreasing long-term flow of timber and carbon accumulation. The development of national forest resources and forestry was simulated for the next century using the Kupolis decision support system and assuming that current forest management is continued under the condition of three scenarios, differing by climate change mitigation efforts. Potential development trends of key forest attributes were analysed and compared with projected carbon stock changes over time, incorporating major forest carbon pools—biomass, harvested wood products and emission savings due to energy and product substitution. The key finding was that the total carbon balance should remain positive in Lithuania during the next one hundred years; however, it might start to decrease after several decades, with steadily increasing harvesting and a reduced increase of forest productivity. Additionally, incorporating the harvested wood and $CO_2$ emissions savings in carbon balance evaluations is essential.

**Keywords:** sustainable forest management; evenness of timber flow; carbon stock changes; simulation; climate change

## 1. Introduction

Sustainable forest management in the broad sense considers economic, ecological and social forestry objectives as being equally important [1] meaning that, in addition to long-term even wood production, forests and forestry play an important role in delivering other ecosystem services [2]. Aiming to secure and increase diverse forest ecosystem services, forestry usually faces conflicts, but also synergies, in relation to forest management [3–5]. Even though there are theories stating that sustainable wood production automatically secures the desirable delivery of other forest ecosystem services [6,7], this has been challenged by numerous studies. There are reports suggesting that under sustainable forest management, there may be almost no reduction or even some gain in biodiversity with an increase in wood production [4,5,8,9]; however, the prevailing opinion is that the intensified forestry required for wood production is responsible for habitat loss and the declining delivery of certain ecosystem services [10–12]. The role of forests and forestry for sequestering carbon and reducing greenhouse gas emissions is essential [13]. The carbon sequestration capacity of forests is also linked to management regimes at single tree, forest stand, or landscape levels [5,14–17].

According to the most recent statistics, the share of the Lithuanian forest sector in terms of the total national value added was 4.6% in 2017 [18], thus making it an important

contributor to the national economy. Current forestry in Lithuania can be characterised as struggling between "the new" and "the old", i.e., rapid social changes are pushing for reforms in forestry, while forestry at large is trying to nurture the traditional professional values, norms and structures [19,20]. It has experienced several land reforms, including a complete ban of private forest ownership and forestland privatization, including restitution to former landowners and their heirs during the last century [18,21,22]. Period of rather intensive forest utilization before the WWII [22,23] was followed by the aspiration to preserve domestic forest resources during the Soviet period and the forest harvesting more than doubled since the restoration of the independence in 1990 due to the transition to a market economy [20,24]. The forest land proportion reached the lowest ever value (~26%) in the current-day territory of Lithuania just atter the WWII [25], however, large-scale afforestation projects, as well as the natural transformation of other land areas to forest resulted in permanently growing forest land area afterwards [19,24]. Nevertheless, the dependence of Lithuanian forestry on command-and-control forest governance, which, among others, involves strict following of the guidelines and relying on "classical" forest management concepts and on the opinions of weighty experts, remains strong [26].

The theory of normal forests, born in Germany and matured in Russia, continues to shape current Lithuanian forest management and alternatives, suggesting that more focus is placed on sustained profits or deliveries of other ecosystem services, are usually scarcely discussed in scientific publications [5,19,27,28]. This theory emphasises the sustainable (in a narrow sustainability context) delivery of timber from forests of possibly the highest productivity [27]. To ensure the evenness of timber flow, forest management is planned and implemented with the aim of achieving an even forest age class distribution. Very long rotations are applied in Lithuania [28], focusing on maximising the deliveries of the highest possible amount of timber of large sawlog dimensions. This aim is legally binding as it is incorporated into the forestry legislation, e.g., the Forest Felling Rules [29] require the Optina model (see Materials and Methods) to be used in practically all state-owned forests for estimating final harvesting budgets, which is an implementation of classic German normal forest principles [30]. No forest stand is allowed to grow in age beyond the desired rotation age in a normal forest, theoretically. Nevertheless, the forests in Lithuania are aging, e.g., commercial state-owned pine forests are felled by final harvesting at an average of 14 years above the minimum allowable harvesting age [18], which is reported to be above the economic rotation age or even average value of technical rotation age if soil productivity is taken into consideration [28]. This number is 17 for spruce forests, 16 for birch, 19 for black alder, and 30 years for aspen. The harvesting/increment ratio in commercial state forests was reported to be 68% and in private forests, it was 65% [18], suggesting deviation (underharvesting) from sustainable timber deliveries. Therefore, the areas of forests at ages beyond the minimum allowable final harvesting age are increasing, with the potential for an even faster increase in the near future, as the post-WWII afforestation is approaching maturity [28]. A number of studies exploring the future of Lithuanian forestry suggest a high potential and even necessity to intensify forestry in terms of increased forest harvesting [28,31,32], thus, the load on sustainability of a multitude of delivered forest ecosystem services may only increase, unless the current forest management's aims and principles are not carefully checked.

It is obvious that forest management also has a significant impact on carbon sequestration and climate change mitigation, which is itself becoming an increasingly important goal [33]. Forest ecosystems have a strong mitigation potential. Therefore, carbon management is essential for climate change mitigation [34,35]. Carbon management in forests is basically aimed at enhancing carbon sequestration in various parts of the ecosystems and the long-term carbon conservation in biomass or wood products [36–41]. Forests contribute to the global carbon cycle via $CO_2$ uptake from the atmosphere in a growing stock volume and $CO_2$ (as well as other greenhouse gases (GHGs)) emissions due to mortality, (wild)fires and the decay of wood products. It is evident that forests contribute to climate change mitigation not only via carbon sequestration, but also substitution—replacing fossil-based

products and fuels for both material and energy production. Even though forest, through carbon sequestration and substitution of fossil-based resources, contributes significantly to remove $CO_2$ from atmosphere, the role that forests should play in mitigating climate change and how it should be accounted, is still widely debated [42]. There have been increasing intentions to use wood for energy purposes with the assumption that bioenergy from forests is climate neutral and emissions should not be accounted [42–47]. At the same time, some studies argue that biomass used for energy production is not carbon neutral [48–53]. The opinion of wood used for bioenergy being carbon neutral is supported by the use of "displacement factors", which are applied for evaluations of GHG emissions vs. carbon stock contained in wood [54]. The carbon neutrality of wood used as a bioenergy source is defined via the assumption that all $CO_2$ emissions due to the combustion of wood are offset by the $CO_2$ captured for the growth of trees, but additional harvesting implies a change in the forest carbon stock and forest availability to sequester $CO_2$ and ensure neutralization [54]. Debates on using wood, whether carbon neutral or not, call for more studies evaluating the footprint of biogenic carbon in wood used for solid and energy production.

Probably due to the abovementioned disagreements and the lack of widely agreed methodology to estimate the displacement factors, substitution effect has not been included in GHG reporting and accounting, while there is a clear and exact methodology for carbon sequestration and conservation in living biomass, dead organic matter and wood products [55] and this is annually reported in countries' national GHG inventories. Currently, the first accounting rules adopted in EU [56] has included forests only as a flexibility/compensation option for other GHG emitting sectors' (not included in Emission Trading System) to the limited extent [57]. This flexibility to cover part of other sectors' emissions is granted only if "no debit" rule—total emissions from land use and forestry sector shall not exceed removals from the same sector—is ensured. For forestry part it means that compensation could be used if carbon sequestration in forest land is larger compared to the reference level set by Land Use, Land Use Change and Forestry (LULUCF) Regulation [56]. The substitution effect may be significant in Lithuania, taking into account that biofuel has the largest share and potential among other renewable fuel sources [58]. The importance to include substitution effect into the GHG accounting is expected to increase even more in the future, since the annually reported carbon sequestration in Lithuania has gradually been decreasing since 2012 [59]. With decreasing carbon sequestration in the above-mentioned pools, the country's efforts to mitigate climate change with carbon conservation in long-use harvested wood products and reduce GHG emissions with bioenergy development are becoming critical. Automatically, we face the question of whether Lithuanian forestry is capable of coping with the challenges to cover GHG emissions from other sectors via carbon sequestration. Are the fundamental forestry principles born two centuries ago able to meet the needs of future generations?

International agreements not only require annual reports on GHG emissions and removals, but also future GHG emission and removal projections and schedule policies for achieving GHG emission reduction targets [60]. In order to provide relevant future GHG emissions and reductions, forest management should be taken into consideration, especially in order to provide results for different scenarios, applying different policies and measures. This requires information on future projections of forests and forestry, which may be achieved by applying decision support systems. Modern decision support systems are capable of modelling the development of forest resources and delivery of various ecosystem services [5,61,62]. Future projections of carbon sequestration using decision support systems are becoming a conventional approach for fulfilling international agreements and commitments and in scientific research [5,8,9,63]. Therefore, to experiment with Lithuanian forestry development in the future and estimate its ability to ensure continuous carbon sequestration, we used the functionality of the decision support system.

The first question asked in this simulation study addresses whether current forest management principles in Lithuania guarantee the non-decreasing timber delivery in a

long run. Then, we test whether aiming for sustainability in timber delivery secures non-decreasing carbon stock changes in Lithuanian forests. For that we project long-term carbon sequestration in Lithuanian forests splitting it into compounding major forest carbon pools, i.e., the aboveground and belowground biomass, wood products and emissions savings due to substitution. Finally, we end with a discussion and proposals for forest and carbon management policies. We hypothesise that that the total carbon sequestration should remain positive in Lithuania during the next one hundred years; however, it might start to decrease after several decades, with steadily increasing harvesting and a reduced increase of forest productivity.

## 2. Materials and Methods

First, the development and use of Lithuanian forest resources were modelled, assuming that (i) current forest management practices are continued and (ii) specifying alternative future scenarios, related to different human efforts to mitigate climate change and, therefore, setting different conditions for forest management. The modelling covered the next 100 years, i.e., the period from 2020 until 2120. Then, using the information on the modelled forest conditions every 10 years for the whole period and the timber harvesting projections during each particular decade, we estimated the total carbon balance (carbon stock change) in Lithuanian forests, splitting it into compounding major forest carbon pools, i.e., (i) carbon stocks in forest aboveground and belowground biomass, (ii) carbon stocks in harvested wood products (HWP) and (iii) $CO_2$ emissions savings due to energy substitution and product substitution. Finally, we evaluated the potential of Lithuanian forests to absorb the carbon during the next century, linking current forest conditions and sustainable timber delivery objectives to the carbon sequestration outcomes and discussing related forest policy implications, emphasizing sustainability aspects.

### 2.1. Study Area

In this study we deal with forests and forestry in Lithuania—a country situated in Central Europe with central coordinates of 55°10′ N, 23°39′ E with a total land area of 65,200 km² (Figure 1). Lithuanian forests are classified as belonging to the European hemi-boreal mixed broadleaved-coniferous forest type in the transitional zone between the boreal coniferous and the nemoral broadleaved forests [64–66]. According to the official Lithuanian forestry statistics [18], forest covers 33.7% of the country's area. Scots pine dominates (34.5%) in Lithuanian forests, with other tree species occurring with lower proportions—birch (22%), Norway spruce (21%), black alder (7.8%), softwood deciduous (~11.4%) and hardwood deciduous (~3.3%). The average growing stock volume in all forests is 260 m³/ha (263 m³/ha in the state and 258 m³/ha in other forests), and the average age is 54 years (56 years in the state and 52 years in other forests) [18].

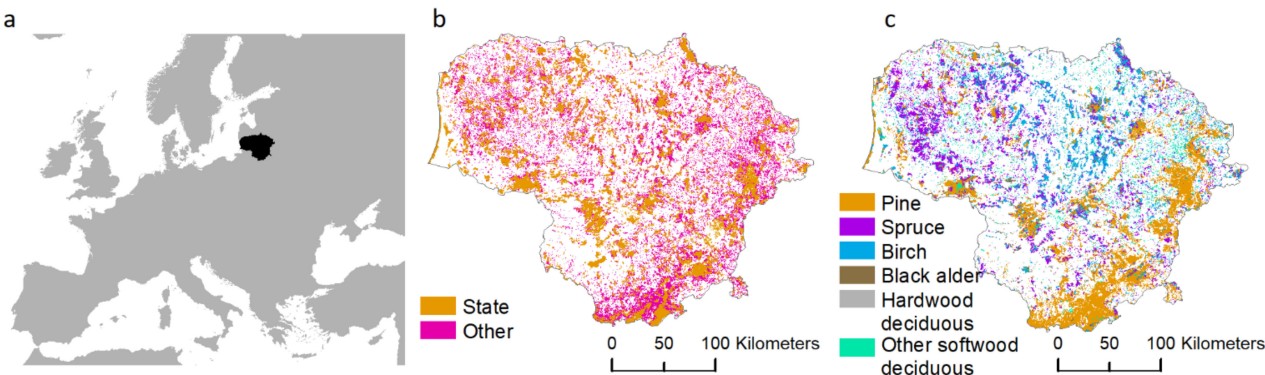

**Figure 1.** Location of the study area and forests in Lithuania: (**a**) Location of Lithuania in Europe; (**b**) forest ownership; (**c**) the most common tree species in a forest block. Sources of spatial data used in the background: (**a**) https://thematicmapping. org/downloads/world_borders.php, accessed on 3 March 2021 and (**b**,**c**) forest state cadastre [67].

### 2.2. Modelling the Development of Forests and Forestry

The development of Lithuanian forests under changing climate conditions was modelled using the Kupolis simulator [68]—a nationally developed simulator that aims to model the development and use of forest resources under conditions of specified management regimes. It is a forest stand level simulator, covering key aspects of forest and forestry development, including forest regeneration, growth, silvicultural treatments, natural mortality, and harvesting. Stand growth projections are based on regression models developed for eight dominant tree species. For each species and each stand, the same models are used to estimate mean gross annual increment and its components including wood left in the stand as the result of management thinning and self-thinning mortality. Recently, the simulator was enhanced to account for the impacts of climate change on forest growth (see below). The final harvesting module in Kupolis is based on the model Optina (abbreviated from Lithuanian "optimal use") which is specified in the Forest Felling Rules [29]. Optina is an implementation of cutting budget estimation approaches based on the theory of normal forests and aimed at continuous and non-decreasing timber deliveries, smoothing of age class structure, and balance between cutting and increment at forest management unit level [69]. It estimates five cutting budgets securing even areas of final harvesting during the whole rotation and 1–4 decades and adjusts the minimum achieved value depending on the average age and age class area distribution for each tree species. Forest estate was considered as the forest management unit. There were 42 management units used for state forests, corresponding to state forest enterprises as they were in 2016. The annual budget of final cuttings is re-optimized at each step using the principles of dynamic programming, while other forest management activities are modelled using iterative simulations. More information on the Kupolis simulator is available in the literature [4,8,62]. The basic input data for Kupolis correspond to the contents and structure of data available from stand-wise forest inventories and are stored in the forest state cadastre, which covers all forests of the country [67]. The output includes two data sets: (i) A detailed description of forest states after each simulation step, identical to the contents of information available from stand-wise forest inventories for every forest stand, and (ii) the lists of forest stands with at least one tree harvested due to any reason, with identification of this reason during the period of each simulation step. Identical forest stand attributes are available in both data sets, providing the information required to estimate the carbon sequestration, however, outside the Kupolis simulator.

In this study, we used data on all forest stands in the country available from stand-wise forest inventories carried out in Lithuanian forests during 2006–2016. The total area of forest stands in 2016 was 2.058 million ha, which slightly fluctuated over time due to forest development and forestry activities. In further analyses, we split all Lithuanian forests into two categories by forest ownership: the state-owned forests (1.044 million ha in 2016) and private and reserved for restitution forests, hereafter referred to as other forests (1.014 million ha in 2016). We did not assume that the total area of forest land, i.e., the areas allocated for forestry activities, changed over time; however, we did assume that the forest management practices remained the same as they were in Lithuania in the current decade no matter the ownership—they were appropriately specified in Kupolis and are fully compatible with other recent studies [70–72]. The only difference in simulation in state-owned and other forests was the number of years specified to harvest out available resources of mature and overmature forests. It averaged 15 years in state forests, while other forests (basically the private ones) were strived to be harvested immediately after passing the minimum allowable final harvesting age limit. The forest management was specified according to the so-called forest group each forest stand was allocated to depending on prioritised forest function. The Forest Law of the Republic of Lithuania assumes four forest groups [73]. Group 1 forests (1.6% of forest land in our study) are unmanaged strict reserves with no human intervention. Natural forest development was simulated in Group 1 forest. Non-clear harvesting only was simulated in stands allocated to Group 2 forests (12.4%), i.e., the ecosystem protection and recreational forests. The rotation age was here close to

natural tree mortality age. Rotation ages in protective or Group 3 forests (14.4%), aimed at timber deliveries with emphasis on protection of soil and water, were prolonged by ten years, if compared with the commercial or Group 4 forests (71.6%). Commercial forests are managed to ensure stable wood supply. The minimum allowable final harvesting age used in simulations was specified according to Forest Felling Rules [29], see Figures A1 and A2 for more information final harvesting age for selected tree species.

The stand-wise forest inventory used to be implemented in Lithuania using 10-year cycles, meaning that each forest compartment was revisited every 10 years. Therefore, the age of information for different areas could differ. Therefore, we simulated the forest conditions in the year 2016 in the first run using appropriate forest management and climate change mitigation scenario specifications. Then, we conducted the simulations for 2020–2120 in 10-year steps.

### 2.3. Accounting for Changing Climate Conditions

To account for climate change effects on forests and forestry and, therefore, influences on the forest growth, timber delivery and carbon sequestration, we used scenarios developed by the International Institute for Applied Systems Analysis (IIASA) for H2020 project ALTERFOR [74–76]. The three scenarios were as follows:

(1) Reference, which assumed the continuation of climate change mitigation efforts in the EU as they were in 2016 and small efforts globally, resulting in strong climate change and a temperature increase of ca. 3.7 °C by 2100 compared to pre-industrial values. Globally, this was expected to increase timber and pulpwood harvests with relatively unchanged proportions and a medium level of logging residue extraction. In Lithuania, this scenario assumed the relatively largest increase of the yield of coniferous forests in Lithuania.

(2) EU bioenergy, which assumed strong EU climate change mitigation efforts and medium efforts globally, therefore resulting in medium climate change and a temperature increase of ca. 2.5 °C by 2100 compared to pre-industrial values. Globally, this was considered to result in increasing pulpwood shares until 2050 and, after that, a strong increase of both timber and pulpwood harvests, with logging residues extraction, as for the reference scenario. The productivity of coniferous forests in Lithuania was projected to increase, however, less than under conditions of the reference scenario.

(3) Global bioenergy, which assumed strong climate change mitigation, thus resulting in halted climate change with a temperature increase of ca. 1.5–2 °C by 2100 compared to pre-industrial values. This was characterised by high bioenergy demand increases, especially for the harvests of small-diameter pulpwood. All available residues were considered to be extracted for energy purposes. The productivity of coniferous stands was projected to increase, however, less so than under the EU bioenergy scenario.

The growth models originally implemented in Kupolis were developed using empirical data collected more than three decades ago. Therefore, the forest growth parameters used in Kupolis were modified to account for climate change in our scenarios. The assumptions on changes in forest growth followed Augustaitis et al. [77], who suggested adjusting the diameter, height, and stem volume increment due to increased temperatures. On average, the stem volume increment was increased by maximum 39%, 30% and 25% for spruce, 21%, 16% and 13% for pine, and 4%, 3% and 2% for deciduous tree species in 2120, assuming Reference, EU bioenergy and Global bioenergy scenarios, respectively. All other forestry principles remained unchanged under different scenarios, due to our focus on current forest management practices and low adaptiveness of Lithuanian forestry. So, the scenarios were directly associated with changed growth by tree species, resulting in changes of tree dimensions and then leading to changed relevant stand characteristics.

### 2.4. Assessed Attributes Describing Forests and Timber Delivery

To describe the development trends of Lithuanian forest resources and timber supply, we picked out the following variables, available from the simulations:

- Standing volume (m$^3$/ha)—gross remaining wood volume of the living trees in the forest stands after harvest and mortality in the preceding decade.
- Area-weighted average age (years).
- Annual net wood volume increment per 1 ha (*IV*) of total forest stand area (m$^3$/ha/year), estimated as:

$$IV = \frac{(SV_{t_1} - SV_t + HV_{t \, to \, t_1} + MV_{t \, to \, t_1})}{10} \tag{1}$$

  where *SV* is the standing volume; *HV* is the harvest volume; *MV* is the mortality volume; *t* and $t_1$ are two subsequent points of time (e.g., *t*: 2020 and $t_1$: 2030). To get the mortality rates, we used data on volumes of trees that died, were harvested, and remained growing from permanent sample plots of the Lithuanian national forest inventory (NFI) referring to three 5-year inventory cycles in 1998–2016 [78].

- Volume of annually harvested timber per 1 ha of total forest stand area (m$^3$/ha/year). The distribution of harvested timber by assortments (sawlogs, pulpwood, logs remaining in the forest and harvest residues) and their prices were taken from the state forestry statistics referring to the years from 2013 to 2018 [79]. To reduce the number of assortment categories, the pulpwood volume also included the volume of roundwood for particle board and firewood.

- Total area of forest stands, belonging to a specific age class (ha). As an age class, we assumed 10-year-long forest stand age intervals, i.e., the 1st age class ranging from 1 to 10 years, the 2nd age class from 11 to 20 and so on.

The age-class structure is a determinant of the forest resource development and forestry and the Lithuanian forest policy stresses the objective of pursuing an even age-class structure, i.e., aiming for equal areas of all age classes within a management unit for specific tree species. As an indicator to quantify deviation of the age-class structure from the targeted equal age class area distribution at a certain time, we used the so-called *K* index [69,72]:

$$K = 1 + \frac{1}{N} \sum_{i=1}^{N} \frac{F_i - F_T}{F_T \times 10^{(i-1)}}, \tag{2}$$

where $F_i$ is the area of the *i*th age class; *i* is the ranking number of the *i*th age class, starting from the mature forests, i.e., having just reached the minimum allowable final harvesting age according to Forest Felling Rules [29], all mature and overmature (age exceeds the minimum allowable final harvesting age by 30 years, or 20 years in soft-wood deciduous dominated stands) forests are considered to belong to the same age class, for which the *i* = 1; *N* is the number of age classes in the rotation, as specified in the Forest Felling Rules [29]; and $F_T$ is the target area of the age class aiming for an equal area distribution by age class, i.e., total area of forest stands divided by *N*.

The value of *K* equal to 1.0 indicates a perfectly equal distribution of age classes. *K* values exceeding 1 suggest that the share of mature and premature forests is too high, while values below 1 indicate the dominance of young and middle-aged stands in the landscape.

### 2.5. Assessing Carbon Sequestration

Approaches and software tools in the statistical environment R developed by P. Biber and K. Black within the frames of H2020 project ALTERFOR were used for calculations of carbon sequestration and carbon balancing [9,80]. As the input in carbon calculations, we used detailed information on forest conditions and harvesting over time, available from the forest simulations. Such information included the volumes of growing stock after each simulation step, proportion of deciduous trees, annual increments, volumes of natural mortality, and harvested timber during the preceding decade, distributed by the main

assortments (sawlogs, pulpwood, logs remaining in the forest for some reason, and harvest residues). Three major carbon pools were considered:

(1) Carbon stocks in the forest, including the above- and belowground living tree biomass and deadwood (harvesting residues, stumps and dead roots). The methodology employed for assessing the carbon changes in biomass pools was based on Tier 1 gain-loss method described in the Intergovernmental Panel on Climate Change (IPCC) Guidelines for National Greenhouse Gas Inventories [55]. For that we estimated the above- and belowground biomass gains, losses due to harvest and mortality and transfer of carbon from biomass to harvest or deadwood pools. The aboveground biomass gains were estimated using the wood volume increment per 1 ha data and wood density, biomass to carbon conversion factor and biomass conversion factors from merchantable wood to total biomass adopted for gain-loss method [55]. Total biomass was achieved adjusting the aboveground biomass values using the values of relative share of root biomass in the total tree biomass. Conversion factors used were adopted for deciduous trees and conifers. Biomass losses due to harvest and mortality were derived using the volumes of harvested timber and mortality, available from the simulations. Deadwood carbon stocks included dead logs (harvest residue logs and mortality logs) and roots (including stumps) and they were calculated to decline with time using an exponential first-order decay function and half-life values for dead coarse roots and stumps and aboveground deadwood.

(2) Wood usage and wood products. The carbon stock changes were based on the carbon stored in wood products coming from timber harvesting during each simulation step. The calculations were implemented using an exponential first-order decay model (function) and half-life values for HWP semi-finished products. HWP stock changes were estimated for each semi-finished wood category and using data on wood product inputs and historical HWP data for initial values of carbon stocks in semi-finished wood products. The wood inflow from the harvest into semi-finished woods products was based on harvest residue loss, wood used for production of energy and harvested timber by assortment (available from the simulations), taking into consideration wood getting lost during processing and the shares of assortments allocation to semi-finished product categories.

(3) $CO_2$ emissions savings due to energy substitution and product substitution. Displacement factors (DF) were used to estimate the emission savings. In our study, we considered three basic fossil fuels being replaced and the displacement factor (for calculating substitution of fossil C-emissions) for energetic wood use was based on average values for gas (0.19), oil (0.26), and coal (0.36).

The values of parameters used for calculations are provided in Table 1. If default values available in [55] were not used, they were elaborated based on the expert knowledge of specialists from the State Forest Service, which is responsible for carbon accounting and reporting in the Land Use, Land-Use Change and Forestry (LULUCF) sector in Lithuania.

**Table 1.** The parameter values used in the carbon evaluation.

| Parameter Name | Value |
|---|---|
| Conversion factor of one biomass mass unit to one carbon mass unit | 0.5 |
| Conversion factor from aboveground merchantable broadleaf wood to total aboveground biomass (from $m^3$ into tons) | 0.8 |
| Conversion factor from aboveground merchantable conifer wood to total aboveground biomass (from $m^3$ into tons) | 0.7 |
| Wood density of deciduous trees ($t/m^3$) | 0.47 |
| Wood density of conifers ($t/m^3$) | 0.41 |
| Relative share of deciduous root biomass in total tree biomass | 0.19 |
| Relative share of coniferous root biomass in total tree biomass | 0.26 |
| Relative share of stump volume in the harvest residues' volume | 0.5 |
| Relative share of stumps in aboveground mortality tree volume | 0.1 |
| Half lifetime of dead coarse roots and stumps | 17.5 |
| Half lifetime of aboveground deadwood | 12.5 |
| Relative loss factor from above- and belowground deadwood pools | 0.15 |
| Relative amount of deadwood (related to remaining growing stock), used for initialization of deadwood stocks | 0.1 |

**Table 1.** *Cont.*

| Parameter Name | Value |
|---|---|
| Relative share of the harvested biomass getting lost during processing | 0.5 |
| Relative share of sawlogs being used for production of energy | 0.39 |
| Relative share of pulpwood being processed into wood-based products | 0.56 |
| Relative share of pulpwood being used for energy production | 0.04 |
| Relative share of harvest residues not remaining in the forest, but being used for energy provision or other short-life purposes | 0.3 |
| Initial value of carbon stocks in semi-finished wood products for sawlogs (tC/ha) | 3.96332865 |
| Initial value of carbon stocks in semi-finished wood products for wood-based products (tC/ha) | 1.12310721 |
| Initial value of carbon stocks in semi-finished wood products for paper/pulp (tC/ha) | 0.00428203 |
| Half lifetime of paper | 2 |
| Half lifetime of sawn wood | 35 |
| Half lifetime of wood-based products | 25 |
| Displacement factor (for calculating substitution of fossil C-emissions) for energetic wood use | 0.27 |
| Product displacement factor for wood-based products | 0.47 |
| Product displacement factor for sawn wood | 0.54 |

The structure of our study is summarized in Figure 2.

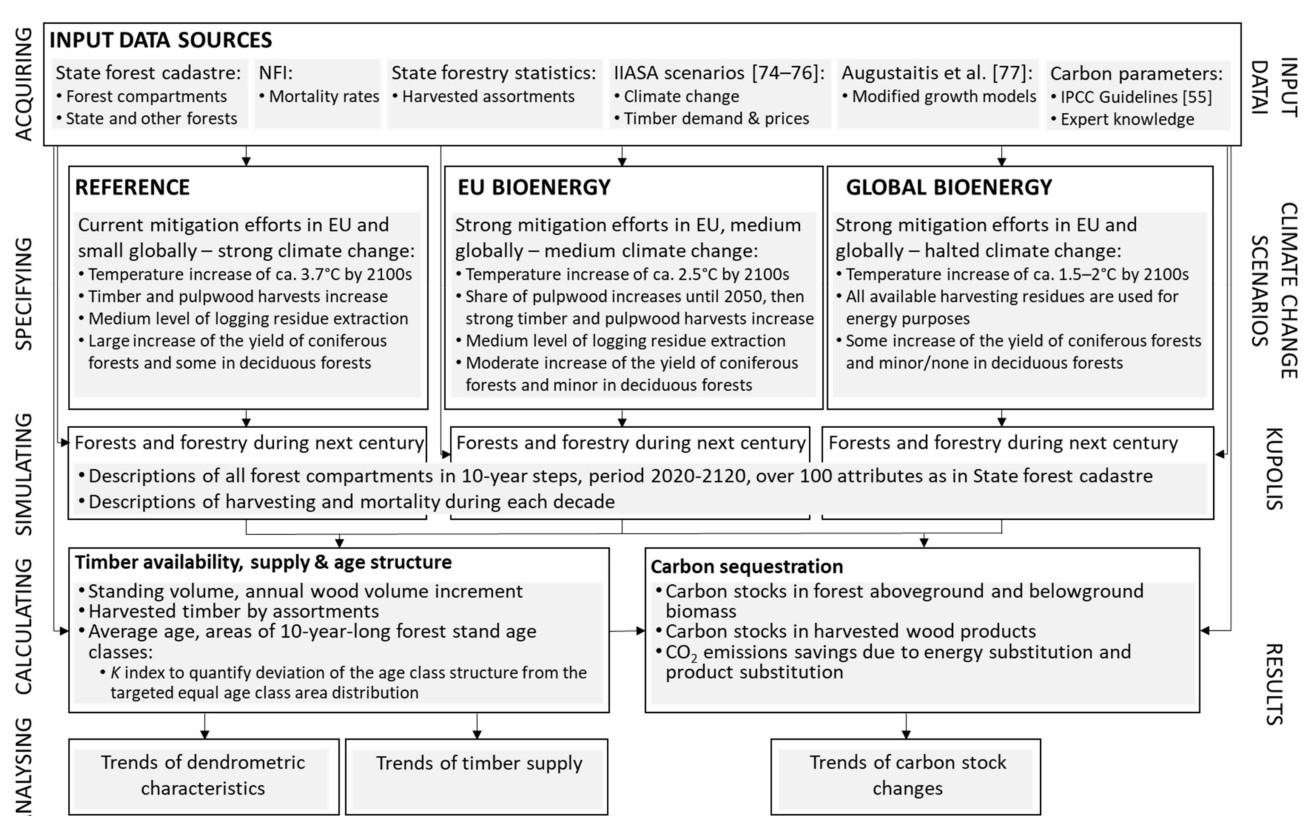

**Figure 2.** Flowchart summarizing overall structure of the study.

## 3. Results

### 3.1. Standing Volume, Age and Timber Harvesting in Lithuanian Forests during the Next Century

The projected development of selected attributes of Lithuanian forests during the coming one hundred years is summarised in Figure 3. Values for all forests are also split into components to separately illustrate the trends in state-owned and other (predominantly private) forests.

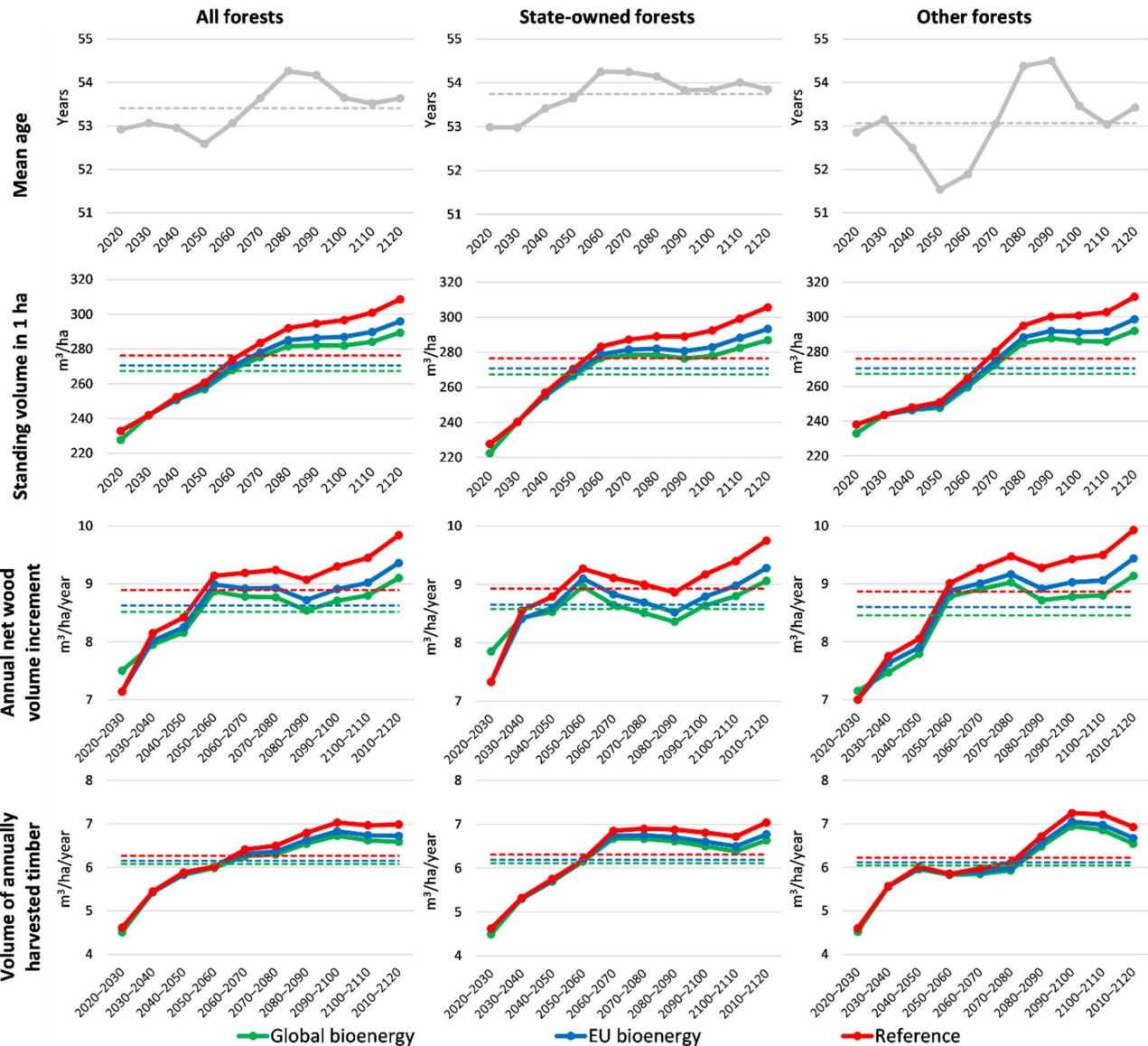

**Figure 3.** Development of selected characteristics of Lithuanian forests and forestry during the next century, depending on the four climate change mitigation scenarios. Dashed lines indicate the average value for each scenario. The mean age development did not differ among the scenarios.

The mean age of Lithuanian forests was projected to remain around 53–54 years during the next century, assuming that there are no major changes in forest management practices. The standing volume in Lithuanian forests, if taking into consideration potentially increasing productivity due to a warmer climate, was projected to increase during the whole period covered. Climate change mitigation scenarios, assuming less human efforts, usually resulted in higher volumes of the growing stock. The annual wood volume increment was suggested to increase rapidly during the next four decades. After this, it was projected to decrease or remain stable in state forests until 2090, before starting to increase again. In other forests, the increment was projected to keep increasing over the whole period, with the exception of the 2080–2090 decade, which is also characterised by the beginning of a jump in the harvesting intensity. Assuming that current forest management principles are continued, the volume of annually harvested timber should increase during the whole period analysed. As a rule, scenarios with higher annual temperatures in 2100 resulted in a higher volume of harvested timber. Harvesting in state-owned forests was projected to grow steadily until 2070, and then remain stable or even decrease until the last decade

covered by our simulations. Harvesting in other forests was projected to increase until 2050 and then from 2080 to 2100.

Even though the objective of Lithuanian forestry has been declared to be aiming for equal areas of age classes through the rotation, the *K* index remained above 1 in practically all cases analysed (Figure 4), suggesting the dominance of relatively more aged forest. Assuming that current forest management practices were continued, the resources of relatively more aged pine forests would accumulate during the first half of the simulated period—until the year 2060 in state-owned forests and 2070 in other commercial (Group 4, following the terminology accepted in Lithuanian forestry [18]) forests. After this, the *K* index would decrease; however, it would not reach a value of 1. The trends for pine in commercial forests were very much followed in protective (Group 3) forests, with just one decade being delayed. The development of the *K* index for other prevailing tree species indicated different trajectories—a decrease of the K value until 2060–2070 in commercial forests and 2070–2080 in protective forests, and then rapid increase for two–three decades, before becoming more or less stable. Some accumulation of old black alder stands in other forests at the beginning of the simulation period should also be mentioned. More detailed information about the areas of age classes during the next one hundred years is presented in the Appendix A (Figures A1 and A2).

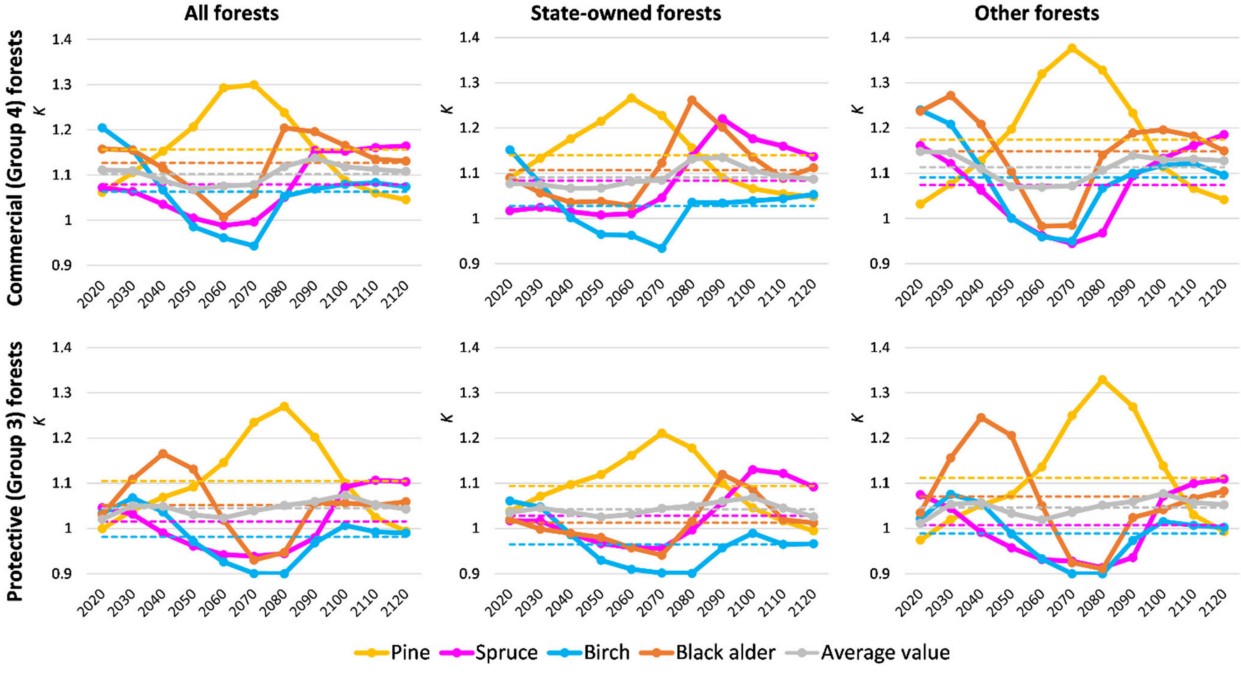

**Figure 4.** Dynamics of the evenness of the age class distribution (*K* index) in forests with different prevailing tree species and management regimes. The grey line indicates the area weighted average of the four tree species analysed.

### 3.2. Carbon Stock Changes in Lithuanian Forests during the Next Century

Assuming that current forest management principles are continued during the next one hundred years, no matter the climate change mitigation efforts, the total carbon balance should remain positive, suggesting that Lithuanian forests will continue to increase the volumes of accumulated carbon (Figure 5). Different climate change mitigation scenarios resulted in outputs which were very much followed in all cases analysed. Relatively, the largest volumes of annual carbon stocks per hectare were achieved under the conditions of the reference scenario, which assumed the least efforts to reduce climate warming and, therefore, under Lithuanian conditions, an increasing productivity of coniferous forests dominating in the country. Both scenarios which assumed increased climate change mitigation efforts (global bioenergy and EU bioenergy) resulted in similar trends of the total carbon balance, with slightly lower values for the global bioenergy scenario with

the most ambitious climate change mitigation targets. The total carbon balance in all Lithuanian forests is expected to increase by 2060. After this, the balance is expected to decrease and continue to decrease until 2110. The balance in all forests started to increase again during the last decade covered by the simulations. If taking into the consideration state forests only, the total carbon stock changes should continue to increase to reach the overall highest values (about 1 tC/ha/year) by 2040. Then, they should start to decrease until 2080–2090, reaching the lowest values (~0.5 tC/ha/year), before starting to increase until the end of the simulation period. The decrease was temporally stopped during 2050–2060. The trends of the total carbon balance in state-owned forests are very much followed in other forests; however, with some delay of breakpoints. The total carbon balance in other forests was projected to increase until 2070–2080, with some reduction during 2040–2050. Bearing in mind similar areas of state and other (predominantly private) forests in Lithuania, in the decade 2060–2070, the other forests would contribute more to the carbon sequestration than the state forests and would keep this status until the end of the current century. The other forests would never reach the maximum total carbon balance values per ha, as achieved by state-owned forests. Additionally, the total carbon balance in other forests never became as low as in the state forests. Except for the current decade, the total carbon balance in all and state forests of Lithuania should remain above the average value for the whole simulated period during the next half-century. Rapidly increasing total carbon balance values in other forests meant that they started to exceed the average values for the whole century for 2050–2060 and maintained this until 2080–2090, under conditions of reducing carbon accumulation potential. There were two peaks projected in terms of the total carbon balance values for all Lithuanian forests—2030–2040, related to the overall maximum of carbon accumulation potential in state forests, and 2050–2060, when the carbon accumulation potential of state forests was still high and that in other forests was approaching its maximum values. The overall lowest carbon balance values were projected for the beginning of the next century, corresponding to the depression in carbon accumulation in other forests.

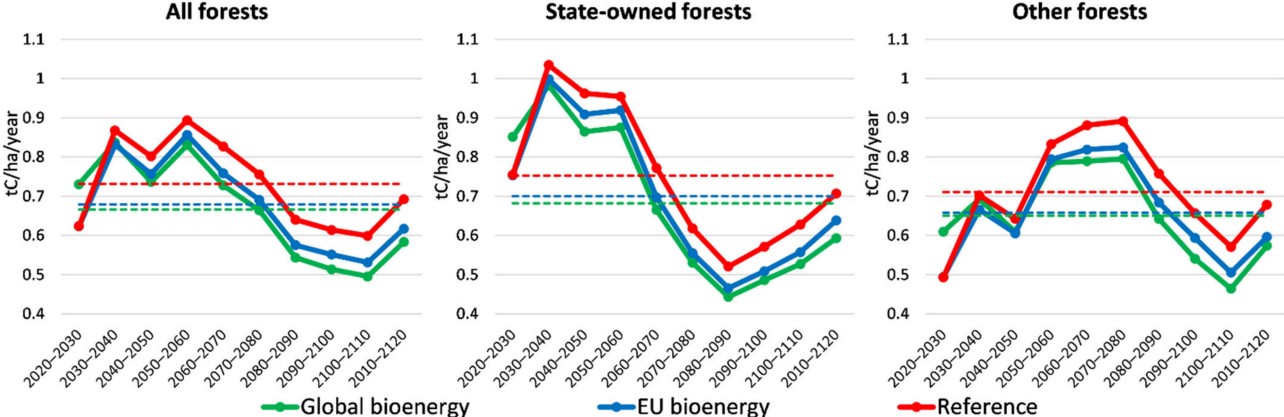

**Figure 5.** Projected total carbon balance in Lithuanian forests during the next century, depending on climate change mitigation scenarios. Dashed lines indicate the average value for each of the four scenarios.

The values of the total carbon balance in Lithuanian forests were split into compounding major forest carbon pools (Figure 6). The trend of carbon stocks in forest aboveground and belowground biomass very much copied the development of the total carbon balance. Even though the average value of carbon stocks in forest biomass was around 0.2 tC/ha/year for all climate change scenarios and forest ownership groups, the were some depressions approaching the turning point at which forest biomass became the carbon emission source. The development of two other carbon pools was projected to be much smoother over time and less varying under conditions of different climate change mitigation scenarios. Carbon accumulation in harvested wood products in state forests remained

rather stable and above the average value for the whole period until 2060–2070, and then started to decrease. Carbon accumulation in wood products harvested in non-state forests was projected to fluctuate over time, with peaks at around 2030–2040 and 2090–2100. Combined carbon stocks in harvested wood products in all Lithuanian forests were projected to increase until 2030–2040, before decreasing to approach 0 by the end of the simulation period. The carbon stocks in wood products harvested in state-owned forests started to increase again during the last decade simulated. Finally, $CO_2$ emissions savings due to energy substitution and product substitution were projected to increase during the whole simulation period. The increase in state-owned forests was steeper before 2060–2070, and then became relatively stable. There were two shorter and steeper increase periods for other forests—from 2020 until 2040 and from 2070 until 2100. If taking into consideration the average values over the whole simulation period, this carbon pool seemed to result in the largest contribution to the overall carbon balance, depending on the climate change mitigation scenario, exceeding the carbon conservation in harvested wood products by more than three times and forest biomass by 1.4–1.8 times.

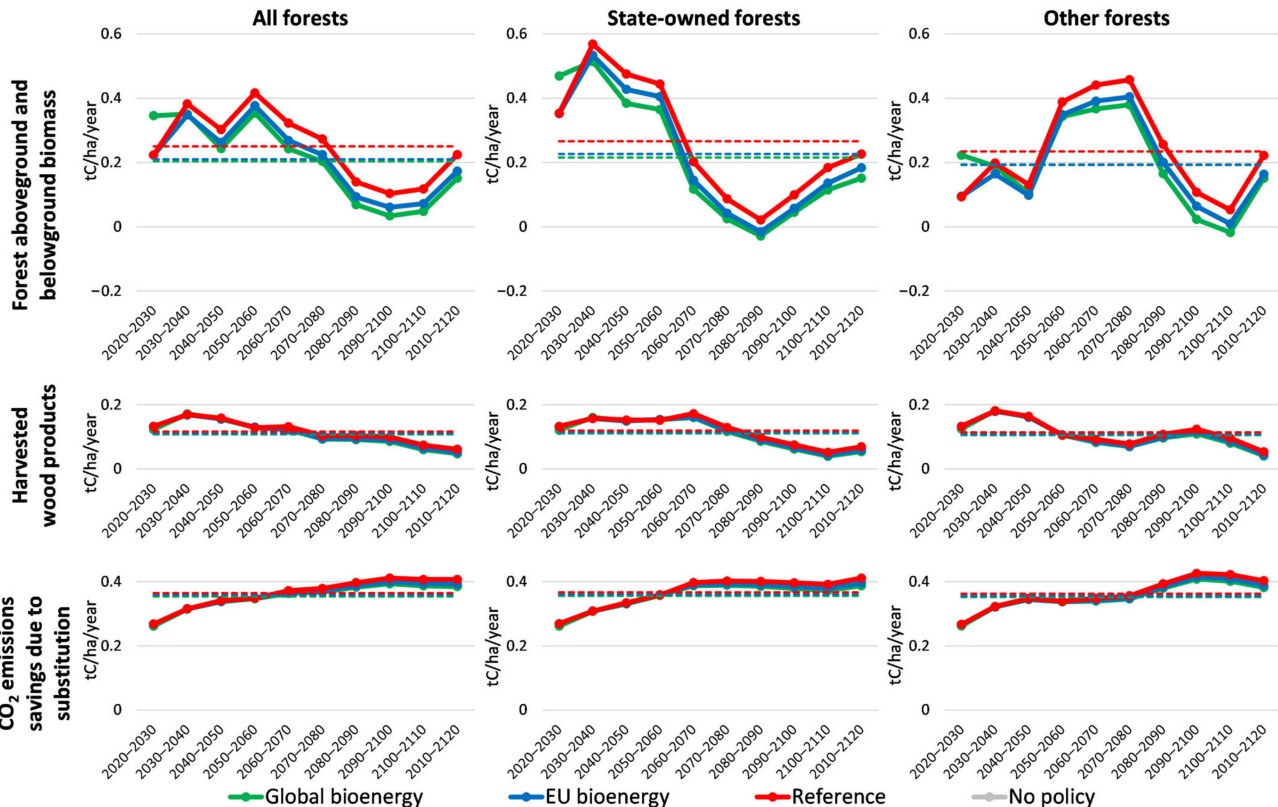

**Figure 6.** Projected carbon sequestration in Lithuanian forests in major forest carbon pools, depending on climate change mitigation scenarios. Dashed lines indicate the average value for each scenario.

## 4. Discussion

The key finding of the current study was the assumption that non-decreasing timber deliveries would be facilitated under warmer climate conditions, assuming that current forest management principles continue to shape Lithuanian forestry in the future. Even though the forest management planning in Lithuania has long been based on the aspiration to achieve an even age class structure, the accumulation of mature and overmature forests, i.e., aged above the minimum allowable final harvesting age, was found to remain an important attribute of national forestry. The accumulation of old pine forests during the next half-century stood out from other observed trends. The projected total carbon balance in Lithuanian forests remained positive during the coming century; however, with a notable

reduction of carbon accumulation during the second half of the simulation period. Carbon stock changes in forest aboveground and belowground biomass also started to decrease during the second half of the simulation period. Carbon stock changes in harvested wood products and emissions savings due to energy substitution and product substitution were important factors when considering the total carbon balance; however, their development trends were much smoother than those for carbon in biomass. Contrary to the increasing trajectories of harvested timber volumes, the accumulation of carbon in HWP decreased during the whole simulation period; however, it remained positive and small, depending on the climate change mitigation scenario. Finally, the $CO_2$ emissions savings due to substitution always increased. State forests were projected to lose their status of the greatest contributor to the carbon accumulation in Lithuania approx. five decades from now. We will analyse these findings in more detail, together with a discussion on forest and carbon management policy implications.

### 4.1. Sustainability of Timber Deliveries from Lithuanian Forests during the Next Century

It seems that the objective of Lithuanian forestry to ensure sustainable timber delivery will be fulfilled during practically the whole coming century if current forest management principles are continued—the volumes of harvested timber only started to decrease when approaching the end of the simulation period. The importance of considering the climate change effects is clearly demonstrated—larger volumes of harvested timber are projected when considering warmer annual temperatures. Of course, this is because of adjustments in forest growth models—we have based our simulations on the assumption that forest productivity in Lithuania in general is increased under warmer temperatures [77]. Nevertheless, consideration of potentially changing forest productivity under changed climate should be mandatory attribute in further forest simulation studies. We did not consider potential disturbing factors, such as windstorms, attacks of pests, and diseases, considering that they have greater impacts in short-term simulations [81]. Nevertheless, upgrade of available forest decision support tools by including explicit representation of disturbances, risk analysis and projecting of potential disturbance effects on forest ecosystems and wood supply chains should be considered future studies. A key assumption was that current forest management practices are continued in the future. This is very much motivated by the objective to draw a reference line, as current management practices in Lithuania are characterised as being very conservative in terms of timber delivery and most other relevant studies have only suggested the intensification of forestry with alternative forest management models [5,28,31,32]. Different climate scenarios were not considered to result in different forest age characteristics, assuming that larger yields and harvests were achieved due to a forest productivity increase under a warmer climate; however, without any impacts on allocating the annual forest felling quota during the simulations, which was done in all cases using the methods of area control and the same minimum allowable final harvesting age [82]. The shape of harvesting trends seems to very much follow that of the mean age. If not taking into consideration the impacts of climate on the forest productivity, the decrease in mean age is usually associated with opposite trends in the annual net wood volume increment, i.e., the increment decreases as the mean age increases. Taking a closer look at the area distribution by age class (exemplified in the Appendix A (Figures A1 and A2)), significant areas of forests older than the 6th age class could be identified (except for commercial state spruce forests with significant shares of young forest), suggesting that relatively older forest would shape Lithuanian forestry, together with the aspiration to "accumulate" the volume and to achieve an even age class structure during the rotation. Considering Lithuanian forest management planning principles, this automatically leads to the reduced harvesting and accumulation of relatively old forests in age beyond the desired rotation age with a potentially lower yield capacity. The risks of current forest management due to the current age class structure of Lithuanian forests were discussed in several previous studies [28,70], noting that large-scale afforestation in the first post-WWII decades predominantly planting pine and the prioritization of spruce

plantations since the 1970s resulted in a large area of forests reaching a final harvesting age in the near future.

The patterns of accumulation of mature and overmature forests (i.e., in age beyond the minimum allowable final harvesting age according to Forest Felling Rules [29]) seem to be dependent on tree species. Here, we analysed just four tree species; however, these cover more than 85% of the Lithuanian forest area. The share of mature and overmature pine forests (according to the trends of the *K* index) should increase until 2060–2080, when post-WWII pine forests are harvested out. The rotation length in commercial pine forests is 110 years and 120 years in protective forests. The changes in the age class structure of forests dominated by other tree species indicated the raising share of relatively younger stands during the first half of the simulation period. The accumulation of mature and overmature forests could be strengthened also by practices applied in Lithuanian state forestry. The methodology applied for estimating the final forest felling quota [82] for an estate using the Optina model (as used in our simulations) requires the quota estimated using the age class area control method and striving for equalizing the areas of all age classes to be further adjusted in a way that the resources of available mature and overmature forests are harvested out, on average, in 15 years, when reaching maturity (note that the planning horizon is 10 years). This automatically introduces additional deviations from sustainability principles in timber deliveries. Common operational practice is that the final forest felling quota, which is legitimized in forest management plans and will become the key factor determining the forest harvesting during the next decade, is determined based on expert opinion or political consensus and it differs from the one estimated to aim for sustainable timber deliveries during the whole rotation. We checked forest management plans elaborated for 11 regional branches of the state company State Forest Enterprise in the period from 2016 [83]. The harvesting intensity during the next decade was adjusted in a way that the number of years to harvest out available resources of mature and overmature forests was changed from an average of 13.6 (standard deviation SD = 1.7) to 18.8 in pine forests, but from 16.3 (SD = 5) to 16.8 (SD = 2.6) in spruce, from 22.4 (SD = 2.8) to 15.6 (SD = 1.1) in birch, and from 24.0 (SD = 7.0) to 16.0 (SD = 1.4) in black alder stands, i.e., the deviations from declared even long-term timber delivery objectives were further increased in pine forest by pushing to accumulate mature and overmature forests. Thus, more intensive use of pine forest resources for timber supply could be discussed.

### 4.2. Carbon Stock Changes in Lithuanian Forests if Current Forest Management Practices Are Continued

Even though the total carbon accumulation in Lithuanian forests was projected to exceed the emissions during the coming 100 years, a decrease of accumulation potential towards the second half of the current century was also observed. These trends were very much followed by the trajectories of carbon stock changes in forest aboveground and belowground biomass, suggesting the links with characteristics of forests and the ways in which they would be managed. General trends of the total carbon balance and carbon stock changes in biomass were in line with the development of the annual net wood volume increment. There was a tendency that, at the beginning of the simulation period, when the gain in increment was still high, the total carbon balance increased. Afterwards, however, the total carbon balance seemed to copy the trends of increment development—stagnation or depression in increment development trajectories was followed by a sudden drop of carbon accumulation in the biomass. The carbon accumulation in biomass usually decreased as the mean age increased. Therefore, the characteristics of available forest resources and the ways in which they are managed are important for securing the sustainability of $CO_2$ sequestration. The importance of changes in forest productivity and harvest levels is also underlined other studies [84–89]. Decrease in carbon accumulation in the HWP pool is related to significant amounts of decaying HWP from the past, resulting in emissions from this pool, therefore increasing input to the pool is not reflected in final HWP stock.

Projected carbon stock accumulation in biomass warns that the potential of carbon accumulation in both Lithuanian state-owned forests and later also in the other forests will

be practically exploited (Figure 6) and even no sequestration is expected for the certain period of time. Such projections call for an immediate action in nowadays forestry to ensure not only continuous forest cover but also continuous carbon sequestration in biomass. One could argue that continuous use of forest resources for energy or solid product could ensure continuous recovery and, therefore, continuous carbon accumulation in forest biomass. However, Global bioenergy and EU bioenergy scenarios both show even lower and no carbon accumulation in biomass at all if compared to the Reference scenario. Even though this is mostly related to differing forest productivity, the use of timber residues use for energy is included as well, therefore harvest is increasing to a lower extent if compared to the Reference scenario (Figure 3). This leads to a proposal to increase the use of forest (timber) resources for both the material and energy production, which may benefit for carbon sequestration and climate mitigation in different ways: (i) via direct $CO_2$ removal from the atmosphere and sequestration in biomass, (ii) carbon accumulated in biomass later preserved in long-term HWP and (iii) GHG emission reduction from other sectors' via substitution (both material and energy source substitution). The importance to evaluate the complete carbon life cycle is stressed also in other studies globally [5,8,44,86,89,90]. Our results support the significance of considering the substitution effect in Lithuanian forestry as it seems to be already contributing more than the carbon preservation in HWP and may overcome carbon sequestration in biomass as well. More, the reduction of GHG emissions due to the substitution is projected to increase significantly—nearly two times—during the next century (Figure 6). Therefore, it may become the most important carbon pool aiming for GHG reduction goals, since biomass carbon sink is projected to reach its maximum sequestration capacity in the end of this century.

Of course, the evaluation of full carbon balance is very sensitive to methodologies used [89]. Should be noted, that we did not consider the emissions impacts of natural disturbances, such as forest fires, carbon stock changes in litter and soils, drainage of organic soils as well as the non-$CO_2$ emissions. To model the carbon dynamics in litter and soil would require detailed inputs and calibration, not available for current study. The default approach of IPCC for carbon stock changes in mineral soils is to consider it as equal to zero for managed forest [55]. We did not consider changes in forest soils to occur. Therefore, the impact of above-mentioned limitations on our findings should be minimal. In any case, we urge to intensify the research in the area of carbon sequestration also in the pools other than forest biomass.

*4.3. Proposals for Carbon Accounting Policies*

Carbon sequestration in this study was estimated splitting it into compounding major forest carbon pools, i.e., the aboveground and belowground biomass, HWP and emissions savings due to substitution. Carbon sequestration in forests as well as emissions are annually reported under the requirements of United Nations Framework Convention on Climate Change [91]; however, only direct carbon stock changes in sinks (biomass, soil, HWP), including emissions from (wild)fires are reported. Taking into account the ambitious goal of the European Union to become the first carbon/climate neutral (or even negative) continent [92], all activities should be accounted for and carbon neutral alternatives for the production of goods and energy are very much needed. Scenarios for climate neutral target achievement in 2050 show two different options: A significant increase of the GHG removals in land use, land-use change, and forestry sector or stable GHG removals in LULUCF sector and an increase in carbon removal technologies [92]. The results of the current study and reports by other researchers [93] show that an increase in carbon sequestration in biomass is temporary and biomass may even turn from a carbon sink to source at a certain period in the future. In addition to this, increasing harvest intensities do not ensure increases of the carbon conservation in the harvested wood products pool, which might be strongly related to the forest management and quality of wood and changes in the distribution of products in groups of different usage periods. Our results provide a solid background for the political discussion of forest management

strategies to be applied in the future for forestry sector to contribute effectively to climate change mitigation, as two different approaches are foreseen for forestry in Lithuania. One is related to the majority of a harvest being used for the production of long-life harvested wood products and carbon conservation in them for decades, while the other is related to increasing the use of lower quality wood from over-mature forests to substitute fossil-based fuels and reduce emissions from energy sector. The reporting and accounting of harvested wood products is clearly described in the legislation [56], but has a limitation for operational use [57]. Emission reduction due to the substitution effect could be seen in other sectors but would result in losses of carbon sink for forestry sector (decreased sequestration or even emissions from biomass).

Current reporting and accounting rules [56] do not include the total carbon balance in forests and thus create uncertainty regarding forestry's role in climate change mitigation. Therefore, the significant potential of forests to mitigate climate change via substitution is neither reported nor projected for the main documents used to define national GHG reduction targets. This is a potential obstacle for policy and measure planning in the forestry sector, since only the carbon sequestration and conservation part of the forestry carbon balance can be accounted for; however, to a limited extent, which is set in the Effort Sharing Regulation [57]. This obstacle and credit limitation from LULUCF sector mentioned before might in turn limit incentives in forestry to adopt carbon sequestration enhancement measures only, since substitution initiatives cannot be evaluated properly. Therefore, the results of this study signal for policy makers to review policies and measures needed to ensure forestry participation in climate change mitigation according to the current legislative acts, since carbon sequestration is projected to decrease both in biomass and HWP in the long term for all scenarios applied. This indicates that the aims of the forestry sector should be discussed in order to plan measures to maintain and enhance forestry potential to contribute in climate change mitigation either via carbon sequestration in forest or via substitution effect.

A decreasing carbon stock change in biomass and harvested wood products calls for changes in forest management to ensure sufficient GHG removals in LULUCF sector, in order to comply with binding and indicative targets. The mandatory "no debit" rule set in the LULUCF Regulation [56] states that GHG emissions from the sector shall not exceed GHG removals in the sector, while the overall goal to reach a climate neutral economy by 2050 indicates that forestry should be able to cover remaining emissions from other sectors. Taking into account the aim to ensure and enhance carbon sequestration in forestry, forest management should aim to harmonize the economic and environmental protection needs for society. While sustainable forest resource use and climate change mitigation should not be opposing goals, environmental protection and biodiversity conservation aims might not always be compatible with carbon sequestration enhancement in forests due to a limited harvest intensity [94,95]; therefore, a balance between protected and actively managed forests needs to be defined.

## 5. Conclusions

Current forest management principles ensure that Lithuanian forests remain a net carbon sink, but do not increase ambitions to mitigate climate change, since carbon sequestration capacity might start to decrease after several decades with steadily increasing timber harvesting and a reduced increase of forest productivity. The importance of the forestry sector could be enhanced if a full carbon balance could be clearly estimated, taking into account forestry's ability to significantly reduce emissions via substitution. The importance of more accurate substitution effect estimation is presupposed for the projected fluctuations in biomass carbon sequestration, as sustainable forest management could not ensure the same carbon sequestration in the long term. Since debates on the eligibility of wood used for energy production for the carbon neutrality criterion are continuing, further studies estimating displacement factors and forest resources needed to capture biogenic GHG emissions from wood are of crucial importance.

Forest decision support tools could be applied in order to estimate the effect of different policy scenarios—aiming to enhance carbon sequestration in biomass or HWP, contribute to the aim of GHG reduction via substitution effect, etc. Forest decision support tools are therefore not only necessary to define the outcomes of the policies planned but also essential in the process to determine the list of policies needed to achieve certain aims. Additionally, both carbon sequestration and substitution effect values are dependent on the share of coniferous and deciduous trees and further division into tree species may also result in a significant difference in final values. We assume that half-life values of harvested wood products might differ in different countries, so the evaluation and application of national half-life values of semi-finished wood products may significantly change the total carbon balance in forests. Another field of study extension is related to roundwood produced and exported to other countries, including how and where the conservation of carbon could be accounted for in products manufactured from that wood. The transition of organic carbon to and from soil during the accumulation of carbon from dead wood and the decomposition of organic matter in soils is another important source for future investigation, taking into account the available results from national studies.

**Author Contributions:** Conceptualization, G.M. and V.K.; methodology, G.M.; software, G.M.; validation, V.K.; formal analysis, G.M. and V.K.; writing—original draft preparation, G.M. and V.K.; writing—review and editing, D.J.; visualization, G.M. All authors have read and agreed to the published version of the manuscript.

**Funding:** This research received no external funding.

**Institutional Review Board Statement:** Not applicable.

**Informed Consent Statement:** Not applicable.

**Data Availability Statement:** Data available on request.

**Conflicts of Interest:** The authors declare no conflict of interest.

# Appendix A

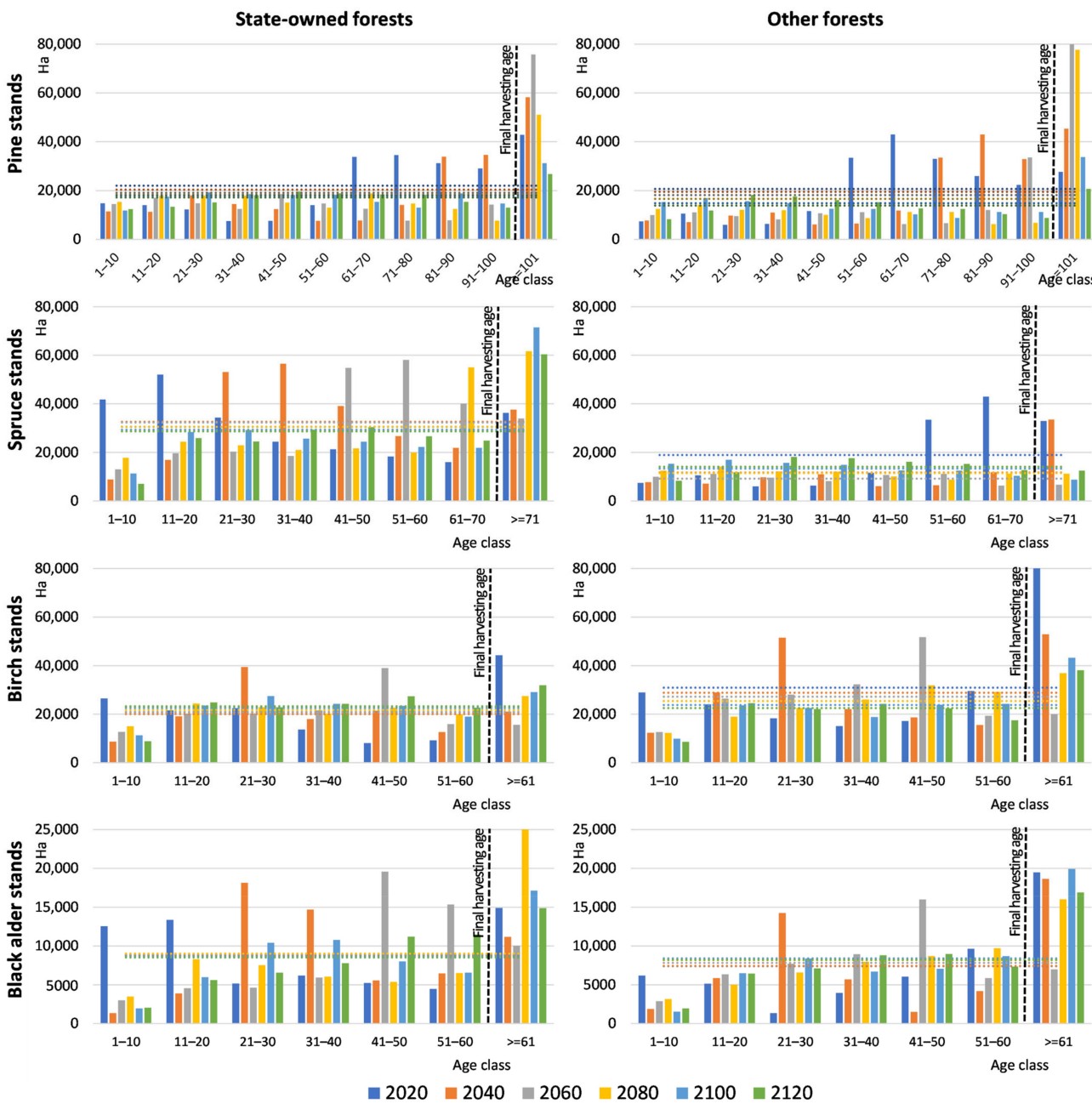

**Figure A1.** Development of the age class structure in Lithuanian commercial (Group 4) forests by prevailing tree species during the next decade. Dotted lines indicate equal areas of all age classes for specific dates targeted by current forest management principles.

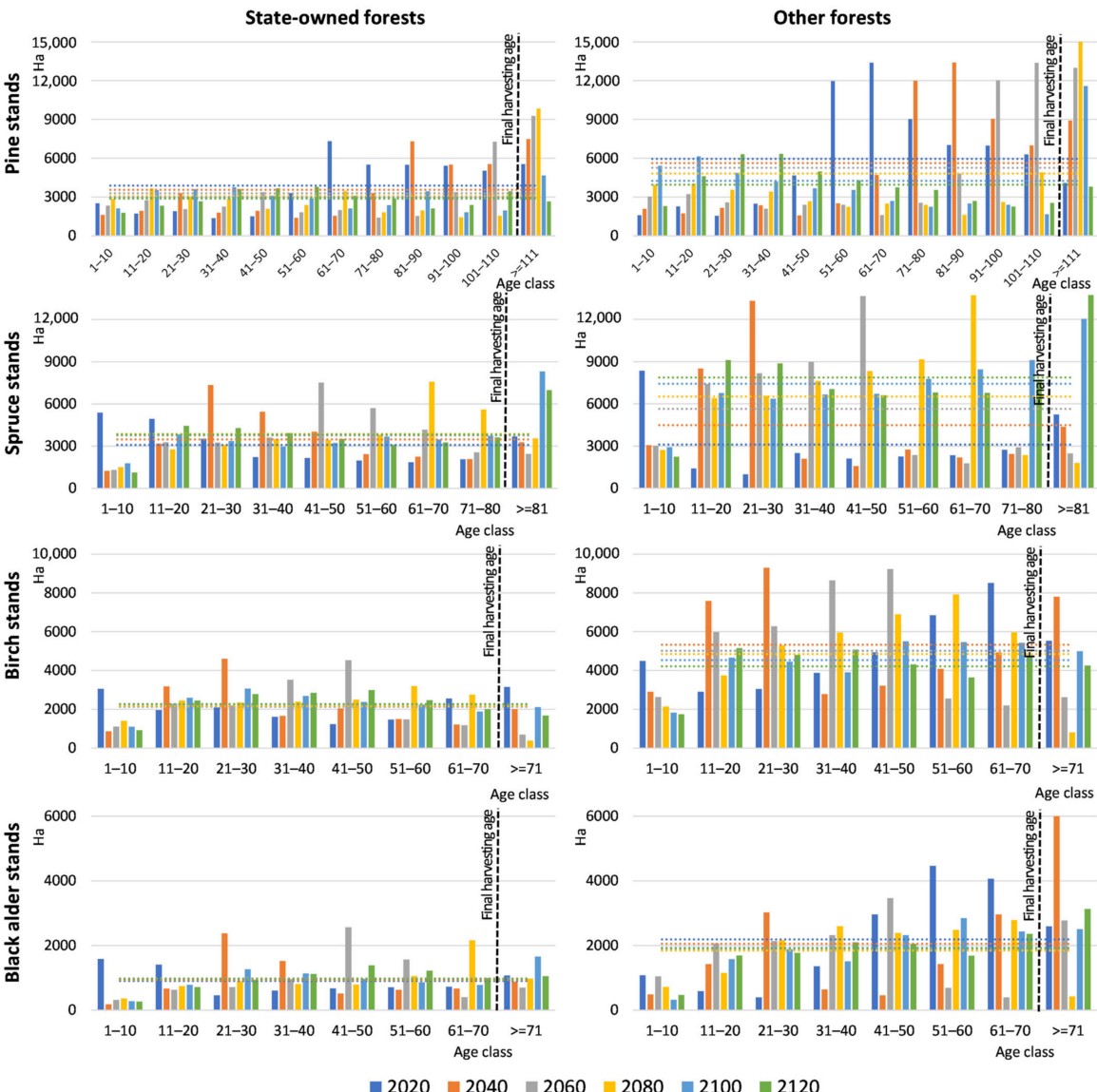

**Figure A2.** Development of the age class structure in Lithuanian protective (Group 3) forests by prevailing tree species during the next decade. Dotted lines indicate equal areas of all age classes for specific dates targeted by current forest management principles.

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
