# Peer review of "Does Aiming for Long-Term Non-Decreasing Flow of Timber Secure Carbon Accumulation: A Lithuanian Forestry Case"

_sustainability, doi:10.3390/su13052778_

Round 1

Reviewer 1 Report

This manuscript looks at different scenarios and their impact on carbon stock changes for Lithuania. While potentially interesting, I think the manuscript needs major revisions before it can be considered for publication. In particular, the methods need to be explained better, keeping in mind international readers that are not familiar with the specifics of the Lithuanian forestry. Unfortunately, I cannot refer to line numbers in my comments and suggestions as the manuscript did not contain any. But I hope you can see what I am referring to nevertheless.

General comments:

You seem to use the word sustainable in two different meanings: in a broad meaning encompassing different various aspects of sustainability (economic, ecological, social), and in a narrow context, as in timber flows that can be sustained over time. I find this very confusing, as it is unclear and open for the readers’ interpretation which meaning of sustainable you mean in different instances in the manuscript. I suggest that you define sustainability and clarify what you mean in the different instances throughout the manuscript so that there is no room for doubt.

You mention biodiversity several times in the introduction and discussion, but it is unclear to me how you consider biodiversity in your methods and results. You should either make that link clear, or shorten the biodiversity-related parts in the introduction & discussion.

How do you deal with climate related risks and measures for adaptation? None of the scenarios seem to assume any adaptation measures?

The discussion contains a rather long elaboration on GHG emissions from the forestry sector. If they are such an important part of the work, I think you should describe the methods used for assessing them a bit more in detail in the methods sections. For example by moving Table A.1 to the methods and indicating which parameters are default, and which are not. Potentially also giving some background on the Lithuanian forestry industry when a parameter differs from the default.

Specific comments:

Introduction

The introduction is quite long and I recommend shortening it if possible, focusing on what is essential.

P2: “poorly managed” – This is an unspecific value statement, can you specify? Poorly in what respect?; “severely depleted” – depleted from what?

What is the Optina model?

P3: ‘which is believed to be too high’ – on what arguments? And too high for what?

‘overmature’: I guess you mean from an economic perspective? Please clarify.

“Simulation studies have predicted a decline of habitat availability for old forest dwelling species due to current forest management practices”: What is the explanation for this? How does it fit with the trend that old forests are increasing?

“products and fuels for both stuff and energy production” replace stuff with material

“sustainable development of forest resources and timber delivery” What meaning of sustainability do you refer to here, broad or narrow? If you mean the broad sense, then I miss an evaluation of more sustainability aspects in your paper. I guess you mean the narrow sense, but please specify.

P4

Your third research question is very unspecific. “..several methodological questions on how to estimate the carbon stock changes..” You should make clear why this is important in the introduction.

“We hypothesise that current forest management principles in Lithuania can secure sustainable timber delivery during the next century; however, forest carbon stock changes may be impacted due to the current characteristics of national forest resources and forestry practices.”

Material and methods

I do not think it is enough to refer to a previous study regarding the tools and input data you have used. You should describe them at least briefly in the manuscript. In particular, I miss information on the following aspects:

  • What is the Kupolis simulator based on – empirical growth and yield functions?
  • How does the Kupolis simulator account for climate change impacts on growth? How confident are you about the growth increase due to climate change, can you compare that to other studies (in the discussion)?
  • Were all forest stands in the whole country inventoried? Why does the forest area fluctuate due to forest development and forestry activities?
  • How did you deal with protected forest and other forest not available for wood supply? What does the differing global demand in the scenarios mean for the Lithuanian case – was this aspect included at all in the simulations, i.e. did you implement different scenarios for wood demand together with different climate change scenarios? If not, why not?
  • The no policy scenario assumes no climate change at all? How can that be compared to the other scenarios if it does not follow the same logic, and why do you include it? You write about this later, but I think the information is needed already in the M&M part.
  • Did you simulate the private forest in the same way as the state forest, or did management prescriptions differ in any way?
  • What are the minimum rotation lengths?
  • You mention in the intro and disc that you use the Optina model, but you don’t explain it in the methods. How does it work and what does it do?
  • Indicators: Why did you choose these indicators?
  • annual net wood volume increment – why is only mortality divided by 10, but not the harvest volume?

Results

General comments: You describe the development of the indicators in quite some detail in the text. I suggest focusing on the most important results in the text, and breaking the text up into more paragraphs to make it easier for the reader. Further, I would not use the word predict in the context of scenario analysis, as long as the scenarios are not meant as some kind of prognosis. I would replace it with e.g. project.

Figure 2: Why do you only show the mean age for the no policy scenario? The fluctuation in age over time is rather small (even though it looks much larger due to cut-offs on the y-axis). Do these small fluctuations really warrant several sentences of text?

Why did climate change result in lower proportions of deciduous species, did your model assume that climate change increases productivity of coniferous species more compared to deciduous species?

What are the different forest groups, they have not been introduced before? (p 9 and Figure 3) And what do you mean with protective forests? I think this needs to be explained in the methods.

Discussion

General comments: Also the discussion would benefit from breaking up the text into several paragraphs.

“Contrary to the increasing trajectories of harvested timber volumes, the accumulation of carbon in HWP decreased during the whole simulation period; however, it remained positive and small, depending on the climate change mitigation scenario.” Why did the HWP pool decrease when harvesting went up?

Your write that: “We did not consider potential disturbing factors, such as windstorms, attacks of pests, and diseases, considering that they have greater impacts in short-term simulations [64]; however, this exercise aimed at supporting strategic or long-term forest planning.”
If disturbances affect in the short term, wouldn't they affect in the long term, too? Especially if risk for disturbances increases in the long term?

“Different climate scenarios were not considered to result in different forest age characteristics, assuming that larger yields and harvests were achieved due to a forest productivity increase under a warmer climate; however, without any impacts on allocating the annual forest felling quota during the simulations, which was done in all cases using the methods of area control and the same minimum allowable final harvesting age”
This needs to be explained already in the methods.

P14 first paragraph: these adaptations need to be explained in the methods. How are these adaptations justified when you earlier state that the scenarios are based on current management?

P14-15: you suddenly start discussing GHG accounting rules. This comes a bit out of the blue; I think you need to introduce these issues already in the introduction.

Author Response

Dear Reviewer,

Thanks for your time to read our manuscript and the valuable comments. We hope, that we managed to adjust the manuscript taking into consideration all of them, even though the time allocated for improvements was rather limited. Much sorry if some of your comments may have slipped our attention.

Please, find below our responses to your comments. Please, note that we changed the title of the manuscript into “Does Aiming for Long-Term Even Flow of Timber Secure Carbon Accumulation: A Lithuanian Forestry Case” to fit better one of your recommendations.

Response to Reviewer 1 Comments

Point 1: This manuscript looks at different scenarios and their impact on carbon stock changes for Lithuania. While potentially interesting, I think the manuscript needs major revisions before it can be considered for publication. In particular, the methods need to be explained better, keeping in mind international readers that are not familiar with the specifics of the Lithuanian forestry. Unfortunately, I cannot refer to line numbers in my comments and suggestions as the manuscript did not contain any. But I hope you can see what I am referring to nevertheless.

Response 1: Thanks for the comments. We tried to follow all recommendations. Special focus was made on specific for Lithuania forestry aspects (like, e.g. definition of mature and overmature forests, methodologies, forest groups) – we explained the meaning when first used or tried to use formulations, hopefully more clear for international reader. This is explained in detail further. We hope we managed to locate all the issues.

Point 2: General comments: 

You seem to use the word sustainable in two different meanings: in a broad meaning encompassing different various aspects of sustainability (economic, ecological, social), and in a narrow context, as in timber flows that can be sustained over time. I find this very confusing, as it is unclear and open for the readers’ interpretation which meaning of sustainable you mean in different instances in the manuscript. I suggest that you define sustainability and clarify what you mean in the different instances throughout the manuscript so that there is no room for doubt.

Response 2: We accept, that this is confusing. Our focus has in principle been on sustainability in a narrow context, e.g. first of all, on even or increasing timber deliveries, etc. Therefore, we checked carefully whole text for using the word sustainability, trying to use this term sparingly, or explain the context. Usually, “sustainability” was replaced by such words like “long-term even flow of …” (of course, where the focus had been on sustainability in a narrow context). We tried to specify where the broad meaning needs to be considered. To avoid confusion, we also changed the title of the manuscript, replacing “sustainable timber delivery” by “long-term even flow of timber”.

Point 3: You mention biodiversity several times in the introduction and discussion, but it is unclear to me how you consider biodiversity in your methods and results. You should either make that link clear, or shorten the biodiversity-related parts in the introduction & discussion.

Response 3: Indeed, the biodiversity related aspects are very weak. The only attribute related to biodiversity was the proportion of deciduous trees in tree species composition. So, we removed biodiversity related results, parts of introduction and discussion, references. The only links to biodiversity remaining in the adjected version of the manuscript were the very general ones, usually supported by references. We also tried very precise with using the term “sustainability” to avoid confusion that we cover all aspects of sustainable forest management.

Point 4: How do you deal with climate related risks and measures for adaptation? None of the scenarios seem to assume any adaptation measures?

Response 4: The study was primarily focused on the assumption, that current forest management principles in Lithuania are continued. In the revised version of the manuscript, we tried to emphasise the low level of adaptiveness of Lithuanian forest management no matter the challenges or opportunities of climate change. We did not report on alternative forest management options, focussing on consequences of continuing with current forestry paradigms. The climate change scenarios used very general, including European and global efforts to mitigate the climate change, and resulting in different timber demands and prices, changed forest productivity. We tried to expand methodology chapter, explaining how the scenarios were implemented in DSS used and limitations due to level of adaptiveness of Lithuanian forestry. We accept that there are numerous aspects not considered in scenarios tested, e.g. like impacts of natural disturbances under changing climate – we tried to note the limitations in the text.

Point 5: The discussion contains a rather long elaboration on GHG emissions from the forestry sector. If they are such an important part of the work, I think you should describe the methods used for assessing them a bit more in detail in the methods sections. For example by moving Table A.1 to the methods and indicating which parameters are default, and which are not. Potentially also giving some background on the Lithuanian forestry industry when a parameter differs from the default.

Response 5: Large part of the discussion on GHG emissions was moved to the Introduction. Methodology chapter was adjusted specifying which of parameters used were based on the default values and which ones were specified by the experts.

Point 6: Specific comments: 

Introduction   

The introduction is quite long and I recommend shortening it if possible, focusing on what is essential.

Response 6: The introduction was revised. We removed some text from the 1st paragraph on the questions related to GHG accounting. The 2nd paragraph was practically completely rewritten replacing long text on the history of Lithuanian forestry that resulted in low adaptiveness with three sentences summarizing the information and, hopefully, delivering the same message. Please, note that some part of information delivered previously in the Discussion, was moved to the introduction, thus, the length of Introduction remaining more the same. We also tried to adjust the Introduction to make it more clear regarding several other aspects, like meaning of sustainability, specific Lithuanian solutions, revised research questions,

Point 7: P2: “poorly managed” – This is an unspecific value statement, can you specify? Poorly in what respect?; “severely depleted” – depleted from what?

Response 7: We used terms originating from the references. We accept that such expressions lack professionalism. However, this part of Introduction was removed from the adjusted version of the manuscript, aiming to reduce the extent and explaining the reasons for evolving current national forestry in few sentences.

Point 8: What is the Optina model?

Response 8: We explained the method for estimating the final felling quota in Materials and Methods. The reference to that was inserted in the Introduction

Point 9: P3: ‘which is believed to be too high’ – on what arguments? And too high for what?

Response 9: We changed the text, stating that “reported to be above the economic rotation age or even average value of technical rotation age if soil productivity is taken into consideration”. We also slightly adjusted other parts of this paragraph, explaining, that the normal forest theory assumes that a forest stand needs to be harvested when it reaches the desired rotation age. In Lithuanian case, the harvesting time usually is under just the physical rotation age.

Point 10: ‘overmature’: I guess you mean from an economic perspective? Please clarify.

Response 10: We tried to replace the word “mature” and “overmature” by more self-explaining text, which is not dependent on Lithuanian understanding, like “… at ages beyond the minimum allowable final harvesting age” and so on. In Materials and Methods, we explained the meaning of “overmature” forests used in Lithuanian forestry, i.e. that this category refers the forest stands which age exceeds the minimum allowable final harvesting age by 30 years, or 20 years in soft-wood deciduous dominated stands

Point 11: “Simulation studies have predicted a decline of habitat availability for old forest dwelling species due to current forest management practices”: What is the explanation for this? How does it fit with the trend that old forests are increasing?

Response 11: We removed the texts related to biodiversity in the adjusted version of the manuscript. Our initial idea was (not relevant to current version of the manuscript) that the habitat availability may go down under current forestry even though the average age and the presence of “overmature” forests increases. I.e. there are other factors in addition to the total areas of old-growth forests which may influence the habitat development.

Point 12: “products and fuels for both stuff and energy production” replace stuff with material

Response 12: Thanks, replaced.

Point 13: “sustainable development of forest resources and timber delivery” What meaning of sustainability do you refer to here, broad or narrow? If you mean the broad sense, then I miss an evaluation of more sustainability aspects in your paper. I guess you mean the narrow sense, but please specify.

Response 13: As explained above, we tried to adjust the manuscript in a way, to underline the “narrow” sense of sustainability. We check the whole text trying to be precise with what are we evaluating. We have chosen just timber supply and carbon sequestration. In principle (not relevant to current version of the manuscript), technically the evaluation of other ecosystem services could be available, however, this would mean in principle elaborating new study.

Point 14: P4  

Your third research question is very unspecific. “..several methodological questions on how to estimate the carbon stock changes..” You should make clear why this is important in the introduction.

Response 14: The last paragraph of the Introduction was essentially revised. The research question related to the carbon stock change estimation methods was rephrased explaining, that we focused on “splitting it [total carbon balance] into compounding major forest carbon pools, i.e. the aboveground and below-ground biomass, wood products and emissions savings due to substitution”. The importance of considering 3 major carbon pools, i.e. accounting also for the substitution effect, was explained in the newly inserted text, i.e. the text we moved in from the Discussion.

Point 15: “We hypothesise that current forest management principles in Lithuania can secure sustainable timber delivery during the next century; however, forest carbon stock changes may be impacted due to the current characteristics of national forest resources and forestry practices.”

Response 15: The hypothesis was changed in the adjusted version of the manuscript: “… the total carbon sequestration should remain positive in Lithuania during the next one hundred years; however, it might start to decrease after several decades, with steadily increasing harvesting and a reduced increase of forest productivity

Point 16: Material and methods        

I do not think it is enough to refer to a previous study regarding the tools and input data you have used. You should describe them at least briefly in the manuscript. In particular, I miss information on the following aspects:           

Response 16: Thanks for “pushing” to provide more information on the methods used. We tried to update this chapter to be independent from other publication. We refer to the publication mentioned in the first version of the manuscript only when discussing the findings. Please, find below more detailed information on the adjustments made

Point 17: What is the Kupolis simulator based on – empirical growth and yield functions?

Response 17: We explained in more details how the forest growth is simulated in Kupolis: “… Stand growth projections are based on regression models developed for eight dominant tree species. For each species and each stand, the same models are used to estimate mean gross annual increment and its components including wood left in the stand as the result of management thinning and self-thinning mortality”.

Point 18: How does the Kupolis simulator account for climate change impacts on growth? How confident are you about the growth increase due to climate change, can you compare that to other studies (in the discussion)?

Response 18: We purely base our expectations regarding the climate change effects on the growth on the study by Augustaitis et al. In fact, this publication is the only available publication related to long term project implemented together with Lithuanian NFI to collect empirical data on retrospection of increment changes using dendrochronological methods. We would also not like to penetrate very much into the arguing climate change effects on forest productivity in Lithuania (which are indeed different from Southern or Central Europe) – this is discussed in other our study, i.e. the one referred to in the initial version of the manuscript. Also – we removed the No policy scenario from current version of the manuscript aiming to focus timber supply and carbon sequestration. I.e. we having fear that more in-depth discussion would lack support by the Results. We inserted a new paragraph in Materials and Methods shortly describing modifications of growth models implemented in Kupolis.

Point 19: Were all forest stands in the whole country inventoried? Why does the forest area fluctuate due to forest development and forestry activities? 

Response 19: Yes. All forest stands, including all private forests, were inventoried and considered in our study. The fluctuations were in the area of forest stands, i.e. the forest land area with tree cover, basically due to clearcuts.

Point 20: How did you deal with protected forest and other forest not available for wood supply? What does the differing global demand in the scenarios mean for the Lithuanian case – was this aspect included at all in the simulations, i.e. did you implement different scenarios for wood demand together with different climate change scenarios? If not, why not?

Response 20: We inserted some text explaining that “The forest management was specified according to the so-called forest group each forest stand was allocated to depending on prioritised forest function…”. This newly inserted text elaborates also on the forest groups, asked about in one of further comments. E.g. we simulated just natural forest development, mortality and no human intervention in Group 1 forests; or natural development with non-clear felling at physical rotation age in Group 2 forests. We did not consider some newly introduced (actually, still in the process of introducing legally) potential habitats of European importance, which may introduce some forest harvesting limitations in commercial forests. We accept that this will have impacts on both timber deliveries and carbon sequestration, however, we believe that general trends will not be altered significantly. We have some simulation results with alternative forest management models, including with notably increased areas under no active management (not relevant to current version of the manuscript), however, not mature for publication yet, also due to unknown legal status of potential habitats of European importance.

Regarding the global demand for timber - this was considered in the climate change scenarios, it resulted different timber prices, then different incomes and profits from forestry activities. However, we suppose, this goes beyond the scope of current paper. The “current forest management” in Lithuania is, indeed, planned and implemented with the aim of achieving an even forest age class distribution, i.e. caring about the age of forest but not the economic reasoning… Therefore the demands were not considered. 

Point 21: The no policy scenario assumes no climate change at all? How can that be compared to the other scenarios if it does not follow the same logic, and why do you include it? You write about this later, but I think the information is needed already in the M&M part.

Response 21: The No policy scenario was removed. We also removed all the text which discusses the potential development under the No Policy scenario.

Point 22: Did you simulate the private forest in the same way as the state forest, or did management prescriptions differ in any way?

Response 22: Yes and no. State and private forests are managed following the same principles and legal requirements. The only difference was calculation of the annual budget of final cuttings. The unit for calculation was the forest estate. The state and private estates differed in “the number of years specified to harvest out available resources of mature and overmature forests. It averaged 15 years in state forests, while other forests (basically the private ones) were strived to be harvested immediately after passing the minimum allowable final harvesting age limit.” This was described in the text.

Point 23: What are the minimum rotation lengths?

Response 23: We adjusted the Figures A1 and A2 to specify the minimum allowable final harvesting age. This was also identified in the text: “The minimum allowable final harvesting age used in simulations was specified according to Forest Felling Rules [29], see Figure A1 and Figure A2 for more information final harvesting age for selected tree species”. The term “minimum rotation length” in Lithuanian forest management planning would mean minimum allowable final harvesting age plus 9. We did not use such term in our manuscript.

Point 24: You mention in the intro and disc that you use the Optina model, but you don’t explain it in the methods. How does it work and what does it do?

Response 24: Even though we did not introduce special paragraph on the Optina “model”, we inserted several sentences explaining how it is implemented: “The final harvesting module in Kupolis is based on the model Optina (abbreviated from Lithuanian “optimal use”) which is specified in the Forest Felling Rules [29]. Optina is an implementation of cutting budget estimation approaches based on the theory of normal forests and aimed at continuous and non-decreasing timber deliveries, smoothing of age class structure, and balance between cutting and increment at forest management unit level [69]. It estimates five cutting budgets securing even areas of final harvesting during the whole rotation and 1–4 decades and adjusts the minimum achieved value depending on the average age and age class area distribution for each tree species. Forest estate was considered as the forest management unit. There were 42 management units used for state forests, corresponding to state forest enterprises as they were in 2016. The annual budget of final cuttings is re-optimized at each step using the principles of dynamic programming, while other forest management activities are modelled using iterative simulations.”

In principle, Optina is a method rather than a model. 

Point 25: Indicators: Why did you choose these indicators?

Response 25: The list of indicators was revised. In adjusted version of the manuscript we use the indicators characterizing state of forests and timber supply. Description of some indicators was also adjusted (assortments, mortality)

Point 26: annual net wood volume increment – why is only mortality divided by 10, but not the harvest volume?

Response 26: Unfortunately, we did not understand this comment well. We got the harvest volume and the mortality for 10 years, then estimated net wood volume increment for 10 years. Annual value was achieved dividing by 10 the sum of standing volume change, harvest and mortality.

Point 27: Results       

General comments: You describe the development of the indicators in quite some detail in the text. I suggest focusing on the most important results in the text, and breaking the text up into more paragraphs to make it easier for the reader. Further, I would not use the word predict in the context of scenario analysis, as long as the scenarios are not meant as some kind of prognosis. I would replace it with e.g. project.

Response 27: The Results chapter was split into subchapters. We replaced the word “predict” by “project” in the Results, as well as in other text. We also tried to reduce the text, basically removing the sentences on No policy scenario. The titles of subchapters are: “3.1. Standing Volume, Age and Timber Harvesting in Lithuanian Forests during the Next Century” and “3.2. Carbon Stock Changes in Lithuanian Forests during the Next Century

Point 28: Figure 2: Why do you only show the mean age for the no policy scenario? The fluctuation in age over time is rather small (even though it looks much larger due to cut-offs on the y-axis). Do these small fluctuations really warrant several sentences of text?

Response 28: The mean age usually does don change very much assuming large number of forest stands contributing. The text explaining the development of mean age was reduced. There is only one line used to illustrate the development of one age, because just current forest management was simulated. Three different climate scenarios were assumed to impact the productivity of forests, with the principles of forest management remaining the same (i.e. using the age class area control method for cutting budget estimation). We specified this is the text.

Point 29: Why did climate change result in lower proportions of deciduous species, did your model assume that climate change increases productivity of coniferous species more compared to deciduous species?

Response 29: Basically, yes. Please, note, that we do not introduce the proportions of deciduous tree species in the adjusted version of the manuscript. Initially, this indicator was considered as a proxy for biodiversity; however, we accepted that our interpretation of biodiversity related aspects without additional indicators and expanding the manuscript would be too weak.

Point 30: What are the different forest groups, they have not been introduced before? (p 9 and Figure 3) And what do you mean with protective forests? I think this needs to be explained in the methods.

Response 30: Thanks for comment. The groups were introduced in the Materials and Methods: “The forest management was specified according to the so-called forest group each forest stand was allocated to depending on prioritised forest function. The Forest Law of the Republic of Lithuania assumes four forest groups [73]. Group 1 forests (1.6% of forest land in our study) are unmanaged strict reserves with no human intervention. Natural forest development was simulated in Group 1 forest. Non-clear harvesting only was simulated in stands allocated to Group 2 forests (12.4%), i.e. the ecosystem protection and recreational forests. The rotation age was here close to natural tree mortality age. Rotation ages in protective or Group 3 forests (14.4%), aimed at timber deliveries with emphasis on protection of soil and water, were prolonged by ten years, if compared with the commercial or Group 4 forests (71.6%). Commercial forests are managed to ensure stable wood supply.”

“Protective forests” is the title for Group 3 forests, used also in several English references.

Point 31: Discussion 

General comments: Also the discussion would benefit from breaking up the text into several paragraphs.

Response 31: The Discussion chapter was split into 3 subchapters, focusing on different research questions:

“Sustainability of Timber Deliveries from Lithuanian Forests during the Next Century”

“Carbon Stock Changes in Lithuanian Forests if Current Forest Management Practices are Continued”

“Proposals for Carbon Accounting Policies”

Point 32: “Contrary to the increasing trajectories of harvested timber volumes, the accumulation of carbon in HWP decreased during the whole simulation period; however, it remained positive and small, depending on the climate change mitigation scenario.” Why did the HWP pool decrease when harvesting went up?

Response 32: We explain this (also, in the text):

Decrease in carbon accumulation in the HWP pool is related to significant amounts of decaying HWP from the past, resulting in emissions from this pool, therefore increasing input to the pool is not reflected in final HWP stock.”

Point 33: Your write that: “We did not consider potential disturbing factors, such as windstorms, attacks of pests, and diseases, considering that they have greater impacts in short-term simulations [64]; however, this exercise aimed at supporting strategic or long-term forest planning.”

If disturbances affect in the short term, wouldn't they affect in the long term, too? Especially if risk for disturbances increases in the long term?

Response 33: We did not predict long term risks of disturbing factors nor their impacts on the forests. We accept this and indicate in the discussion, that: “… upgrade of available forest decision support tools by including explicit representation of disturbances, risk analysis and projecting of potential disturbance effects on forest eco-systems and wood supply chains should be considered future studies”.

(Not relevant to our manuscript) – we have good examples, that e.g. heavy wind damages in Lithuania 10 years ago did not result in notable deviations from strategic figures much due to “resilience” of current forest management in Lithuania. We modelled in our study the resistance of forests to several disturbance types (wind and diseases) under conditions of climate scenarios, however, we did not have reliable tools for analyses of risk development.

Point 34: “Different climate scenarios were not considered to result in different forest age characteristics, assuming that larger yields and harvests were achieved due to a forest productivity increase under a warmer climate; however, without any impacts on allocating the annual forest felling quota during the simulations, which was done in all cases using the methods of area control and the same minimum allowable final harvesting age”. This needs to be explained already in the methods.

Response 34: We hope that we have explained this in the adjusted version of the manuscript. Material and Methods were revised including extra text on the forest management planning principles and the simulations (see above). Only “current forest management principles” were tested in our study based on the methods of area control and the same minimum allowable final harvesting age under all scenarios. There is only one line used to illustrate the age development in the Results, and explained shortly in the adjusted version of the manuscript.

Point 35: P14 first paragraph: these adaptations need to be explained in the methods. How are these adaptations justified when you earlier state that the scenarios are based on current management?

Response 35: Sorry, if we did not understand well the comment. The 1st paragraph on page 14 discusses common operational practice to delay with final harvesting. We added a recommendation that “more intensive use of pine forest resources for timber supply could be discussed”. Also, the methodology chapter has been revised, hopefully covering the concerns by the reviewer.

Point 36: P14-15: you suddenly start discussing GHG accounting rules. This comes a bit out of the blue; I think you need to introduce these issues already in the introduction.

Response 36: The GHG accounting rules, i.e. the issue with to carbon pools to consider, was introduced in the Introduction in the revised version of the manuscript. We also revised the research questions, hopefully introducing more clear reasoning why we focus so much on GHG accounting.

Reviewer 2 Report

The issue addressed in the paper discusses the current forest management principles in Lithuania. The authors analyzed whether the applied principles of forest management can secure the sustainable development of forest resources and delivery of ecosystem services, with a special focus on carbon accumulation. This, of course, was verified on the case of Lithuania. In my opinion, an important and interesting topic was proposed, compatible with the scope of the journal.

Such studies are partially analysed in literature. It would be worth presenting the state of the art in a broader way. State of the art should be more related to your research questions.

Generally, in the proposed scope, the paper was prepared correctly. However, I recommend a few corrections to improve the quality of this article:

1) to precisely define the research scenario (it is very general); needed to clarify the scope of the study and consequently a clear, step-by-step, simple, synthetic research pattern; I recommend more precision, as the reader should know how to repeat a similar analysis on this basis (please consistently correct and complete points 2 and 3); not enough only "is described in more detail in our previous study";

2) to improve the readability and description of figures (since they are the basis for model verification and discussion of K-index values), supplement the history of their description, a clear and not laconic reference in the paper; (please verify the grouped Fig. 1-3, and A1; justify their relationship to the aim of the study and your research questions);

3) to explain briefly whether there is  need to use, for instance, other methods (first of all, I suggest explaining what are the weaknesses of the proposed method, and what are the strengths, practical advantages (please complete point 4);

Should the scientific research be based on various debates on the eligibility of wood used for energy production for the carbon neutrality criterion are continuing? What is the answer to your research question, formulated at the start of the paper? I also strongly suggest that recommendations for specific, practical, not only general (and not entirely clear) applications of this research shall be provided (please complete point 5).

The language of this paper is relatively correct, however some descriptions would benefit from being more concise. I think it is worth using the help of a native speaker.

Author Response

Dear Reviewer,

Thanks for your time to read our manuscript and the valuable comments. We hope, that we managed to adjust the manuscript taking into consideration all of them, even though the time allocated for improvements was rather limited. Much sorry if some of your comments may have slipped our attention.

Please, find below our responses to your comments. 

Please, let us also summarize some updates done following the requests of the 1st reviewer. We changed the title of the manuscript into “Does Aiming for Long-Term Even Flow of Timber Secure Carbon Accumulation: A Lithuanian Forestry Case” to underline that we do not penetrate into sustainability of forestry in a broader sense. We also made major changes in the Introduction. Results now include only facts about basic forest, timber supply and carbon sequestration. Subchapters were introduced both in Results and Discussion.

Response to Reviewer 2 Comments

Point 1: The issue addressed in the paper discusses the current forest management principles in Lithuania. The authors analyzed whether the applied principles of forest management can secure the sustainable development of forest resources and delivery of ecosystem services, with a special focus on carbon accumulation. This, of course, was verified on the case of Lithuania. In my opinion, an important and interesting topic was proposed, compatible with the scope of the journal.

Response 1: Thanks for your exact interpretation of our objectives to address the current forest management principles and issues in Lithuania and the comments and suggestions we used to improve the manuscript. Please, note that some of your comments were overlapping with the ones by another reviewer, so, we could also overlap with our answers.

Point 2: Such studies are partially analysed in literature. It would be worth presenting the state of the art in a broader way. State of the art should be more related to your research questions.

Response 2: We made major revisions in the Introduction. First of all, we reduced significantly the part discussing Lithuanian forestry during the last century and the reasons behind current forest management. We expanded the paragraph discussing the issues related to the normal forest theory and its impacts on operational forestry. We did not penetrate into the principles of this theory. The updated version of the manuscript contains expanded analyses on the importance of forest management on carbon sequestration and the issues with carbon accounting. As one of our research questions is related to which carbon pools should be considered and how they are rerated to current forest management, we additionally discussed the issues on GHG reporting and accounting in the adjusted version of the Introduction.

Point 3: Generally, in the proposed scope, the paper was prepared correctly. However, I recommend a few corrections to improve the quality of this article: 

1) to precisely define the research scenario (it is very general); needed to clarify the scope of the study and consequently a clear, step-by-step, simple, synthetic research pattern; I recommend more precision, as the reader should know how to repeat a similar analysis on this basis (please consistently correct and complete points 2 and 3); not enough only "is described in more detail in our previous study";

Response 3: The chapter on Materials and Methods was revised essentially. We removed references to our previous study and tried to make the descriptions of methods used completely independent from other descriptions. We provided more information on the solutions used to simulate the development of forests and forestry, explaining the functioning and specifications used in DSS Kupolis, its adaptation to deal with different climate change scenarios, the ways the “normal forest theory” is operationally implemented, some specifics of forest management in Lithuania, etc. The descriptions of input and output variables were refined. Following the recommendations of other reviewer, we removed the No policy scenario and some output variables, we used initially as proxy for biodiversity. To respond to your request to define precisely the research scenario, we introduced Figure 2 which is a flowchart summarizing overall structure of the study.

Point 4: 2) to improve the readability and description of figures (since they are the basis for model verification and discussion of K-index values), supplement the history of their description, a clear and not laconic reference in the paper; (please verify the grouped Fig. 1-3, and A1; justify their relationship to the aim of the study and your research questions);

Response 4: Sorry, if we misinterpreted this your request. We revised the research questions, to be clearer and more strongly linked with the results. We introduced subchapters both in Results and Discussion, to be clearer, that usually the 1st subchapter refers to the evenness of long-term timber flows and forest characteristics (Figure 3) and agreement (or deviations) with the assumptions of normal forest theory (like aiming for equal areas of age classes, discussed in Figure 4 and illustrate in more details in the Appendix), while the 2nd subchapter addresses associated carbon sequestration issues.

Point 5: 3) to explain briefly whether there is  need to use, for instance, other methods (first of all, I suggest explaining what are the weaknesses of the proposed method, and what are the strengths, practical advantages (please complete point 4);

Response 5: We updated the discussion identifying the weakness of our solution used. We accept that there are numerous issues related to simulation-based studies. So, we tried to underline some potential methodological gaps, like not accounting for potentially increased risks due to natural disturbances, etc. In the updated version of the manuscript, the focus in the discussion basically was made on the limitations of current forest management on sustainability of timber deliveries and the limitation of current GHG accounting and reporting principles.

Point 6: Should the scientific research be based on various debates on the eligibility of wood used for energy production for the carbon neutrality criterion are continuing? What is the answer to your research question, formulated at the start of the paper? I also strongly suggest that recommendations for specific, practical, not only general (and not entirely clear) applications of this research shall be provided (please complete point 5).

Response 6: We accept and are very sorry that the initial version of the manuscript contained some statements and not clear formulations which were later not supported enough by the results. We edited the Introduction removing some parts and revised the research questions. The Discussion is structured in a way, that each subchapter has a practical recommendation. The 3rd subchapter of the discussion is supposed to be a recommendation on carbon accounting and reporting, based on our calculations and analyses of current carbon accounting issues.

Point 7: The language of this paper is relatively correct, however some descriptions would benefit from being more concise. I think it is worth using the help of a native speaker.

Response 7: Unfortunately, we had no time left for additional language checking. The first version of the manuscript was edited by MDPI language editing services. So we plan to do with the updated version of the manuscript, however, due to limited time allocated to submit the adjusted manuscript, we could not get such service 

Round 2

Reviewer 1 Report

I think the manuscript has improved considerably. I think there would be more room for improvement, but I agree that this is difficult given the short time available to make revisions. If possible, you could clarify the methods regarding the model and the HWP pools a bit more still, and possibly shorten the overall text as it is quite wordy. I don't have concrete suggestions, though.
The harvest level increases quite a lot over time, I am not sure that even flow is the best term, would non-decreasing be an option? Maybe there is another, better alternative.
Regarding the estimation of annual net volume increment on page 8: I think the formula is missing that you also have to divide harvest volume by 10 (not only mortality) to arrive at annual values.
Another minor comment: on p18 L14-15 there is a repetition of the words 'carbon accumulated in...'.

Author Response

Response to Reviewer 1 Comments

Point 1: I think the manuscript has improved considerably. I think there would be more room for improvement, but I agree that this is difficult given the short time available to make revisions. If possible, you could clarify the methods regarding the model and the HWP pools a bit more still, and possibly shorten the overall text as it is quite wordy. I don't have concrete suggestions, though.

Response 1: Thanks for your evaluation. We revised the paragraph in Materials and Methods on “Assessing Carbon Sequestration”. We updated the description of methodologies estimate all carbon pools not only the HWP, but also carbon stocks in the forest and substitution effect. We hope that the newly added text explains the approaches used in our study. We did not provide formulas which are available in references ([55]). We inserted the following text (inserted text is indicated using Italic font):

On the carbon stocks in the forest:

“Carbon stocks in the forest, including the above- and belowground living tree biomass and deadwood (harvesting residues, stumps and dead roots). The methodology employed for as-sessing the carbon changes in biomass pools was based on Tier 1 gain-loss method described in the IPCC Guidelines for National Greenhouse Gas Inventories [55]. For that we estimated the above- and belowground biomass gains, losses due to harvest and mortality and transfer of carbon from biomass to harvest or deadwood pools. The aboveground biomass gains were estimated using the wood volume increment per 1 ha data and wood density, biomass to carbon conversion factor and biomass conversion factors from merchantable wood to total biomass adopted for gain-loss method [55]. Total biomass was achieved adjusting the aboveground biomass values using the values of relative share of root biomass in the total tree biomass. Con-version factors used were adopted for deciduous trees and conifers. Biomass losses due to harvest and mortality were derived using the volumes of harvested timber and mortality, available from the simulations. Deadwood carbon stocks included dead logs (harvest residue logs and mortality logs) and roots (including stumps) and they were calculated to decline with time using an exponential first-order decay function and half-life values for dead coarse roots and stumps and aboveground deadwood”.

On the HWPs:

“The carbon stock changes were based on the carbon stored in wood products coming from timber harvesting during each simulation step. The calculations were implemented using an exponential first-order decay model (function) and half-life values for HWP semi-finished products. HWP stock changes were estimated for each semi-finished wood category and using data on wood product inputs and historical HWP data for initial values of carbon stocks in semi-finished wood products. The wood inflow from the harvest into semi-finished woods products was based on harvest residue loss, wood used for production of energy and harvested timber by assortment (available from the simulations), taking into consideration wood getting lost during processing and the shares of assortments allocation to semi-finished product categories.

On the emission savings:

“CO2 emissions savings due to energy substitution and product substitution. Displacement factors (DF) were used to estimate the emission savings. In our study, we considered three basic fossil fuels being replaced and the displacement factor (for calculating substitution of fossil C-emissions) for energetic wood use was based on average values for gas (0.19), oil (0.26), and coal (0.36).”

We also reduced the paragraph on parameter used in the calculations. The table listing the parameter values used in the carbon evaluation was moved from the Appendix to become Table 1. Sorry, we did not do that in the first round of adjustments. The table was slightly adjusted, inserting some values available in the code of the software only.

We tried to reduce the text (even though the total number of words could remain near the same due to inserted above mentioned text in the Materials and Methods). We reduced the subchapter in Discussion on Proposals for Carbon Accounting Policies. Also, some parts of our conclusions were removed or rewritten to increase the focus on the most important aspects and removing proposals, which could be easily assumed, however, not directly based on the results of our study. Finally, we increased the use of abbreviations, like LULUCF instead of Land Use, Land Use Change and Forestry.

Point 2: The harvest level increases quite a lot over time, I am not sure that even flow is the best term, would non-decreasing be an option? Maybe there is another, better alternative.

Response 2: Well, in principle, the classical normal forest theory emphasises similar harvested volumes over time. There are references using the term “even” in English. However, the practical implementation of this theory in Lithuanian forestry has resulted in increasing harvest levels, with all associated outputs regarding decrease of forest productivity and carbon sequestration. However, we have to accept that there are deviations from the classical approaches in Lithuanian forestry. Some may argue, that of course it is still “on course for normal forest”, so the truly “even” timber harvests may be expected only during next rotations… To avoid misunderstandings, we tried to avoid the term “sustainable timber flows” when making the first revision. However, the “non-decreasing” is really an option. We revised the manuscript trying to use term “non-decreasing timber flow/harvest/supply/etc” when referring to principles of Lithuanian forestry. However, we would like to keep “even” when talking about the classical normal forest theory. Accordingly, the title of manuscript was changed once again: “Does Aiming for Long-Term Non-Decreasing Flow of Timber Secure Carbon Accumulation: A Lithuanian Forestry Case”. We also changed “stable or increasing” into “non-decreasing”, also resulting in adjusted research questions in the introduction.

Point 3: Regarding the estimation of annual net volume increment on page 8: I think the formula is missing that you also have to divide harvest volume by 10 (not only mortality) to arrive at annual values.

Response 3: To avoid misunderstandings, we created a formula using Equation editor. Yes, we estimate the increment for a decade and divide it by 10 to arrive at annual values.

Point 4: Another minor comment: on p18 L14-15 there is a repetition of the words 'carbon accumulated in...'.

Response 4: Corrected, i.e. the repeated words were removed